# Open-World Drone Active Tracking with Goal-Centered Rewards

**Haowei Sun**[1,2,*] **Jinwu Hu**[1,3,*] **Zhirui Zhang**[1,2] **Haoyuan Tian**[1,2] **Xinze Xie**[1,2]
**Yufeng Wang**[1,5] **Xiaohua Xie**[6] **Yun Lin**[7] **Zhuliang Yu**[1,2,†] **Mingkui Tan**[1,4,†]

[1] South China University of Technology, [2] Institute for Super Robotics (Huangpu), [3] Pazhou Laboratory,
[4] Key Laboratory of Big Data and Intelligent Robot, Ministry of Education, [5] Peng Cheng Laboratory,
[6] Sun Yat-sen University, [7] Harbin Engineering University

## Abstract

Drone Visual Active Tracking aims to autonomously follow a target object by controlling the motion system based on visual observations, providing a more practical solution for effective tracking in dynamic environments. However, accurate Drone Visual Active Tracking using reinforcement learning remains challenging due to the absence of a unified benchmark and the complexity of open-world environments with frequent interference. To address these issues, we pioneer a systematic solution. First, we propose **DAT**, the first open-world drone active air-to-ground tracking benchmark. It encompasses 24 city-scale scenes, featuring targets with human-like behaviors and high-fidelity dynamics simulation. DAT also provides a digital twin tool for unlimited scene generation. Additionally, we propose a novel reinforcement learning method called **GC-VAT**, which aims to improve the performance of drone tracking targets in complex scenarios. Specifically, we design a Goal-Centered Reward to provide precise feedback across viewpoints to the agent, enabling it to expand perception and movement range through unrestricted perspectives. Inspired by curriculum learning, we introduce a Curriculum-Based Training strategy that progressively enhances the tracking performance in complex environments. Besides, experiments on simulator and real-world images demonstrate the superior performance of GC-VAT, achieving a Tracking Success Rate of approximately 72% on the simulator. The benchmark and code are available at https://github.com/SHWplus/DAT_Benchmark.

## 1 Introduction

Visual Active Tracking (VAT) aims to autonomously follow a target object by controlling the motion system of the tracker based on visual observations [80, 75]. It is widely used in real-world applications such as drone target tracking and security surveillance [22, 73, 77, 54]. Unlike passive visual tracking [3, 74, 33, 5, 9, 84, 67, 58], which involves proposing a 2D bounding box for the target on a frame-by-frame with a fixed camera pose, VAT actively adjusts the camera position to maintain the target within the field of view. Passive visual tracking often falls short in real-world scenarios due to the highly dynamic nature of most targets. Thus, VAT offers a more practical yet challenging solution for effective tracking in dynamic environments.

Recently, VAT methods have evolved into two main categories: pipeline VAT methods [40, 46, 15] and reinforcement learning-based VAT methods [19, 39, 18, 80]. **Pipeline VAT methods** employ a sequential framework where the visual tracking [32, 4, 62, 33] and control models are connected in series. The tracking model estimates the target position in the input image and the control model

---

*Equal contribution. Email: sunhoward1105@gmail.com, fhujinwu@gmail.com
†Corresponding author. Email: mingkuitan@scut.edu.cn, zlyu@scut.edu.cn

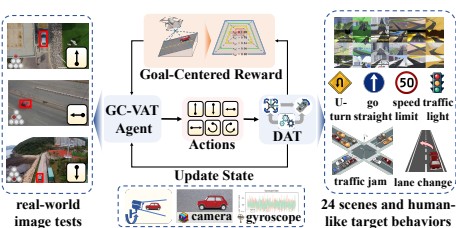

Figure 1: A pipeline for drone VAT.

Table 1: Comparison of DAT benchmark with simulators where existing methods are located.

|  | AD-VAT+ [80] | D-VAT [19] | AOT [39] | **DAT** |
|---|---|---|---|---|
| Scenes | 8 | 4 | 2 | **24** |
| Targets | 1 | 1 | 1 | **24** |
| Tracker | Ground | Drone | Ground | **Both** |
| Dynamics | ✗ | Simplified | ✗ | **Full Physics** |
| Target Behavior | Policy-based | Rule-based | Rule-based | **Human-like** |
| Scene Building | Manual | Manual | Manual | **Digital Twin** |

generates control signals. While this modular design allows for clear task separation, it often requires significant manual effort to label the training data, and the control module requires additional tuning for different scenes. To address these issues, **reinforcement learning-based VAT methods** integrate visual tracking and control within a unified framework. These methods eliminate the need for separate tuning of the tracking and control modules by using a unified framework to map raw visual inputs directly to control actions. Therefore, the reinforcement learning-based VAT methods simplify system design and increase the efficiency of learning adaptive tracking behaviors in dynamic environments.

*Unfortunately*, achieving accurate drone VAT with reinforcement learning remains challenging, partly for the following reasons. **1) Missing unified benchmark.** Existing benchmark scenes are low in complexity, neglect tracker dynamics or rely on overly simplified models, making them inadequate to validate the agent performance (see Table 1). Previous methods [39, 19, 57] use rule-based target management, far from producing human-like target behaviors. Additionally, current 3D scenes are all manually constructed, leading to a heavy workload and limited scene number. **2) Vast environments with complex interference.** Open-world tracking involves large, dynamic environments with frequent interference. In previous methods [39, 19], trackers can only capture images from a fixed horizontal viewpoint. However, the fixed forward viewpoint captures excessive sky, reducing target-related visual information, especially for air-to-ground tracking tasks. Besides, since VAT goal is to keep the target at the image center, such viewpoint restricts the tracker to the same height as the target, severely limiting the perception and movement range. Moreover, training directly in complex conditions leads to slow convergence or difficulty in building strong behaviors.

To address the above limitations, we **first** propose **DAT**, the first open-world active air-to-ground tracking benchmark that simulates real-world complexity (see Fig. 2(b)). Specifically, DAT provides 24 city-scale scenes, full-fidelity simulations of drone dynamics, and a lightweight tool that can be integrated into any 3D scene to enable human-like target behaviors. It also offers a digital twin tool that can generate unlimited 3D scenes from real-world environments, enabling unlimited scene expansion. **Second**, we propose a novel drone VAT with reinforcement learning method (called **GC-VAT**), aiming to improve adaptability in complex and diverse scenarios. Specifically, we design a Goal-Centered Reward to provide precise feedback across viewpoints, enabling the agent to expand perception range through unrestricted perspectives. Besides, we propose qualitative and theoretical methods to analyze the effectiveness of our reward. In addition, inspired by curriculum learning [65, 41, 83], we propose a Curriculum-Based Training strategy that progressively improves agent performance in complex environments. Our contributions are summarized as follows:

1) **A comprehensive drone active tracking benchmark.** We present DAT benchmark, featuring high-fidelity dynamics, 24 city-scale scenes, and tools for simulating human-like target behaviors and unlimited scenes generation, enabling rigorous algorithm validation. 2) **A novel drone active tracking method.** We propose GC-VAT, which leverages a Goal-Centered Reward function and a Curriculum-Based Training strategy to enhance drone tracking performance in complex and dynamic environments. Besides, we propose qualitative and theoretical methods to analyze the effectiveness of our reward. 3) **Extensive experimental validation.** Experiments on simulator and real-world images validate DAT usability and GC-VAT effectiveness, with GC-VAT achieving a Tracking Success Rate of approximately 72% on the simulator.

## 2 Task Definition of Drone Active Tracking

DAT task seeks to train a model to control a drone for active target tracking in dynamic environments (see Fig. 1). Using visual and motion sensor data, the model learns actions to keep the target centered in view, ensuring robust performance across diverse scenarios.

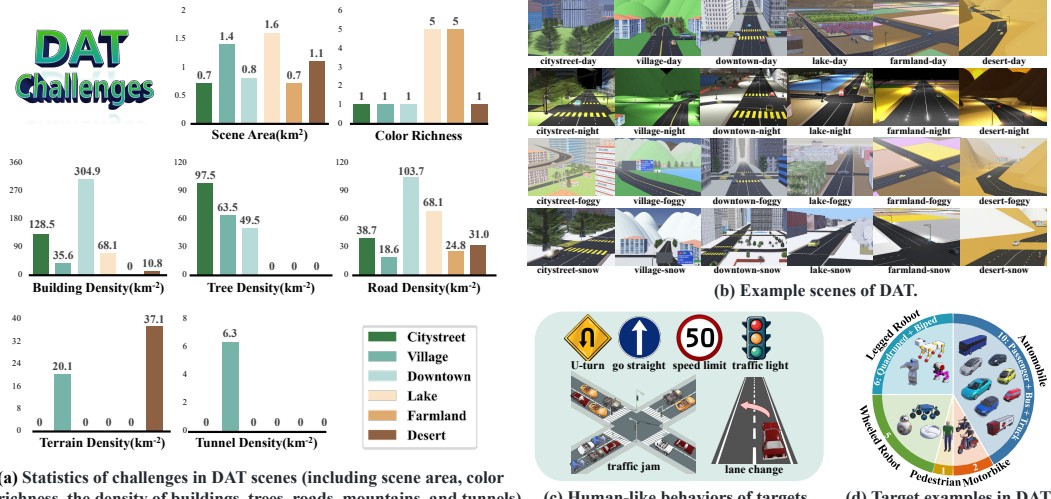

(a) Statistics of challenges in DAT scenes (including scene area, color richness, the density of buildings, trees, roads, mountains, and tunnels).

(b) Example scenes of DAT.

(c) Human-like behaviors of targets.

(d) Target examples in DAT.

Figure 2: Statistics and simulator component examples of DAT. (a) Statistics on 7 complexity aspects in DAT scenes. (b) Example scenes of DAT. (c) Diverse behaviors of targets. (d) Examples of the tracking targets. More details can be found at https://github.com/SHWplus/DAT_Benchmark.

**Observation spaces.** The target is initially positioned at the center of the field of view, and the observation space comprises data acquired from sensors (e.g., RGB images with $84 \times 84$ resolution).

**Action spaces.** The action space can be either discrete or continuous. A discrete space defines a set of predefined drone maneuvers, whereas a continuous space allows direct control over the velocity.

**Success criterion of DAT task.** We define a success criterion when the model can keep the target object, which is initially located at the center of view, in the middle of the image for a long duration.

**Challenges.** Open-world drone active tracking is challenging due to limited data and high risks of trial-and-error in the real world, necessitating complex simulation environments. Additionally, the complexity and dynamics of open-world scenes further demand robust agent performance.

## 3 DAT Benchmark with Diverse Settings

We develop DAT, including 24 city-scale scenes built by an unlimited scene generation tool, high-fidelity drone dynamics simulation, and a versatile pipeline for producing human-like target behaviors.

### 3.1 Diverse Scene Construction

**Digital twin tool.** Users can select any region from *OpenStreetMap* [1] to obtain countless scenes using our tool. Specifically, it generates a high-precision road network with traffic lights and rules, and it converts elevation and vegetation data into 3D assets placed in the scene. Moreover, all assets in the generated scene are editable, allowing for data augmentation. See Appendix B for details.

**Scene construction.** Based on the above tool, we construct 6 outdoor scenes under 4 weather conditions, modeling 7 real-world complexities. Specifically, the *scene area*, *building density*, and *color richness* depict the complexity of the visual background. *Road density* and *terrain density* affect the target behaviors. The *tree density* and *tunnel density* measure the level of visual occlusion. As shown in Fig. 2(a), six scenarios exhibit unique and realistic complexity across the seven aspects:

• **Citystreet scene** covers an area of 0.7 square kilometers. It has a road density of 38.7 and a tree density of 97.5, mainly testing the agent's efficiency against tree occlusions.

• **Village scene** spans 1.4 square kilometers. This scene features a mountain density of 20.1 and a tunnel density of 6.3, requiring the agent to predict the target's movement when it is fully obscured.

• **Downtown scene** covers 0.8 square kilometers. It includes complex road elements and high building density of 304.9, challenging the agent's tracking accuracy and obstacle avoidance abilities.

- **Lake scene** encompasses 1.6 square kilometers. The density of road elements is 68.1, and the richness of background colors is 5, challenging the robustness across varying features and colors.

- **Farmland scene** covers an area of 0.7 square kilometers. The color richness is 5 and multiple color patches, challenging the agent's adaptability to multi-color environments.

- **Desert scene** covers 1.1 square kilometers. It includes a mountain density of 37.1 and a road density of 31.0. Some roads are covered by sand, testing the agent's adaptability to such conditions.

Four weather conditions are designed to test the agent's cross-domain adaptability. **Foggy** reduces visibility, **night** reduces brightness, and **snow** alters the color. The above 24 scenes (see Fig. 2(b)) can fully measure the agent tracking performance. See Appendix B for scene construction details.

## 3.2 Various Trackers and Targets Construction

Drone Active Tracking in the real world involves diverse targets depending on tasks. DAT provides diverse targets with human-like behaviors and enables high-fidelity tracker dynamics simulation.

**Tracker.** DAT benchmark supports two tracker types: drones and ground robots. The drone used is the *DJI Matrice 100* [14], equipped with a *3-axis gimbal*, allowing for precise camera adjustments. Unlike simpler kinematic models in [19] and methods that ignore the dynamics[39], DAT leverages *webots* [37] to simulate the drone's full dynamics, including mass, inertia, aerodynamics, and the response and jitter of the gimbal, closely matching real drones. See Appendix B for details.

**Targets.** DAT includes five categories of targets: *automobile*, *motorbike*, *pedestrian*, *wheel robot*, and *legged robot*, with a total of 24 tracking targets (see Fig. 2(d)). See Appendix B for details.

**Target Management.** We propose a novel pipeline to simulate realistic target behavior. Specifically, DAT first utilizes road networks generated by the tools described in Section 3.1, and directly integrates them with the SUMO traffic simulator [36]. Then, random trajectories are assigned to each vehicle, with SUMO managing its motion. To bridge the gap between simulation and visualization, we implement a controller that translates motion data into human-like driving behaviors for 3D vehicles (see Fig. 2(c)). Our controller also adheres to traffic rules and can simulate phenomena such as traffic light waits and traffic jams. Even better, the controller can be applied to any 3D scene.

# 4 VAT with Reinforcement Learning

In this paper, we primarily focus on visual active tracking (VAT), a core task within DAT benchmark. We propose a drone visual active tracking with reinforcement learning method called Goal-Centered-VAT (**GC-VAT**), aiming to improve the performance of tracking targets in complex scenes. As shown in Fig. 1, we model drone active tracking as a Markov Decision Process (MDP) and train a Drone Agent capable of adapting to unrestricted viewpoint conditions to track a target in the open scene.

## 4.1 MDP for Drone Active Tracking

We seek to learn end-to-end drone tracking policies in dynamic environments by modeling the task as an MDP: $\langle \mathcal{S}, \mathcal{A}, \mathcal{R}, \gamma, \mathcal{T} \rangle$. In this representation, $\mathcal{S}$ denotes the state space, $\mathcal{A}$ represents the action space, and $\gamma$ is the discount factor. At each time step $t$, the agent takes the state $s_t \in \mathcal{S}$ as input and performs an action $a_t \in \mathcal{A}$. Next, the simulator transitions to the next state $s_{t+1} = \mathcal{T}(s_t, a_t)$ and calculates the reward $r_t = \mathcal{R}(s_t, a_t)$ for the current step. The details of the MDP are as follows:

**State $\mathcal{S}$** is the visual information of the scene. At each time step $t$, the camera captures one image of size $84 \times 84$ as the current state.

**Action $\mathcal{A}$** is a set of discrete actions, including *forward*, *backward*, *leftward*, *rightward*, *turn left*, *turn right*, and *stop* movements. At each time step, the Drone Agent selects an action $a_t \in \mathcal{A}$ based on the state $s_t$ and actively controls the camera movement.

**Transition $\mathcal{T}(s_t, a_t)$** is a function $\mathcal{T} : \mathcal{S} \times \mathcal{A} \to \mathcal{S}$ that maps $s_t$ to $s_{t+1}$. In this paper, we use the *webots* dynamics engine to provide a realistic transition function.

**Reward $\mathcal{R}(s_t, a_t)$** is the reward function. The goal-centered rewards are given in Section 4.2.

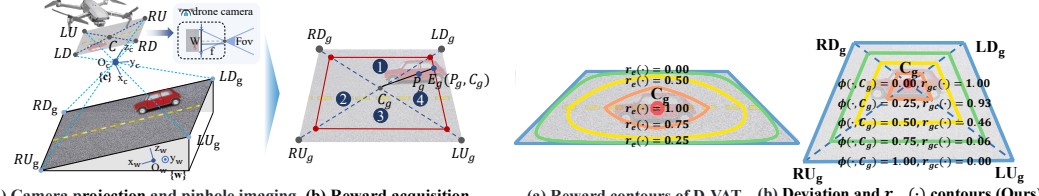

**(a) Camera projection and pinhole imaging** **(b) Reward acquisition**

Figure 3: Diagram of reward acquisition.

**(a) Reward contours of D-VAT** **(b) Deviation and $r_{gc}(\cdot)$ contours (Ours)**

Figure 4: Reward design analysis diagram.

**Network structure of Drone Agent.** Similar to previous works [39, 80], we select a backbone architecture consisting of a CNN followed by a GRU network [11] (see Appendix C.2).

**Key Challenges in Drone Active Tracking.** In open-world environments, drones face unpredictable target behaviors and frequent occlusions. Designing a single reward that encourages diverse and robust tracking actions is extremely difficult. To address this, we propose a **goal-centered reward** in Section 4.2. Moreover, given the vast observation space, discovering successful policies is non-trivial. To facilitate efficient learning, we introduce a **curriculum-based training strategy** in Section 4.4.

## 4.2 Goal-Centered Reward Design

For drone tracking a ground-based target, our reward is designed to characterize the target's position in the image and guide the drone to keep it centered. Therefore, we first need to select an appropriate distance metric to quantify the proximity between the target and the center in the image plane.

Since the drone typically captures images from a top-down perspective, the image plane is not parallel to the ground. Due to the affine transformation, the projection of the image plane becomes a trapezoid (see Fig. 3(b)), and the physical distance between the drone and the target cannot directly correspond to their pixel distance. Existing methods [39, 19] compute the Euclidean distance between the drone and the target, which may not accurately reflect their spatial relationship in the image plane.

To address this issue, we employ a deviation metric $\phi(\cdot, \cdot)$ to measure the distance between the target and the image center projection, as illustrated in Fig. 3(b). Specifically, given a target point $P_g$ and the image center projection $C_g$, the deviation metric is computed by

$$\phi(P_g, C_g) = \frac{|P_g - C_g|}{|E_g(P_g, C_g) - C_g|}, \tag{1}$$

where $|P_g - C_g|$ denotes the distance from $P_g$ to $C_g$ and $|E_g(P_g, C_g) - C_g|$ represents distance from point $E_g(P_g, C_g)$ to the center. $E_g(P_g, C_g)$ is the intersection of the line connecting $P_g$ and $C_g$ with the projected image boundary, as shown in Fig. 3(b).

The deviation $\phi(\cdot, \cdot)$ is designed to ensure that targets inside the image are closer to the center than those outside, with contours shown in Fig. 4(b).

**Principles for Reward Design.** The objective of the VAT task is to keep the target at the image center. Thus, the targets closer to the image center projection should get higher reward values. For deviation metric $\phi(\cdot, \cdot)$, the design principle of the reward function $\mathcal{R}_\phi(\cdot)$ is defined as:

$$\forall P_1, P_2 \in \mathcal{W}, \text{ if } \phi_1 < \phi_2, \text{ then } \mathcal{R}_\phi(\phi_1) > \mathcal{R}_\phi(\phi_2), \tag{2}$$

where $\mathcal{W}$ denotes the valid region with non-zero reward, and $\phi_1$, $\phi_2$ represents the deviation from the target point to the image center projection.

**Goal-Centered Reward Function.** Our reward $r_{gc}(\cdot)$ decreases as the target moves away from the projected image center $C_g$, and is zero if outside, as shown as follows:

$$r_{gc}(P_g) = \begin{cases} \tanh\left(\alpha(1 - \phi(P_g, C_g))^3\right), & P_g \in \mathcal{I}_{clip} \\ 0, & otherwise \end{cases}. \tag{3}$$

The attenuation degree of $r_{gc}(\cdot)$ can be adjusted using the hyperparameter $\alpha$, set to 4. The $\tanh(\cdot)$ provides a strong indication of the task goal due to its relatively quick decay at the image center. $\mathcal{I}_{clip}$ is the truncated image range set to prevent the drone from keeping the target at the edge of the image. The truncation of the image can be controlled using the hyperparameter $\lambda_{clip}$ as: $\lambda_{clip} = H_{\mathcal{I}_{clip}}/H$, where $H$ and $H_{\mathcal{I}_{clip}}$ are the heights of the original and the truncated image. We set $\lambda_{clip} = 0.7$.

**Algorithm 1** Curriculum-Based Training (CBT)

---

1: **Input:** Initial policy parameters $\theta_0$, phase threshold $\eta$, total steps $N$, rollout steps $n$
2: **Initialize:** Training phase $phase \leftarrow 1$, reward buffer $\mathcal{B} \leftarrow \emptyset$, rollout buffer $\mathcal{B}_r \leftarrow \emptyset$
3: **for** each step $k = 0, 1, ..., N-1$ **do**
4:      **if** $phase = 1$ **then**
5:          Configure simple environment: linear target trajectories + no obstacles
6:      **else**
7:          Configure complex environment: varied target movements + obstacles/occlusions
8:      Collect transition $\tau_k = (s_t, s_{t+1}, a_t, r_t)$ with rewards calculated via (3)
9:      Append to buffer: $\mathcal{B} \leftarrow \mathcal{B} \cdot r_k$, $\mathcal{B}_r \leftarrow \mathcal{B}_r \cdot \tau_k$
10:      **if** $k \bmod n = 0$ **then**
11:          Update policy using PPO: $\theta_{k+1} \leftarrow \text{PPO\_Update}(\theta_k, \mathcal{B}_r)$
12:          Clear rollout buffer: $\mathcal{B}_r \leftarrow \emptyset$
13:      **if** $phase = 1$ **and** $\frac{1}{|\mathcal{B}|} \sum_{r_t \in \mathcal{B}} r_t \geq \eta$ **then**
14:          Switch training phase: $phase \leftarrow 2$
15:          Clear buffer: $\mathcal{B} \leftarrow \emptyset$, $\mathcal{B}_r \leftarrow \emptyset$

---

**More details about the Goal-Centered Reward.** The reward function (Eq. 3) relies on the projections of the four corners and image center to compute deviation $\phi(\cdot, \cdot)$. As shown in Fig. 3(a), in the camera frame $\{c\}$, the image center and four corner points have the coordinates $C(-f, 0, 0)$, $LU(-f, -\frac{1}{2}W, \frac{1}{2}H)$, $LD(-f, -\frac{1}{2}W, -\frac{1}{2}H)$, $RU(-f, \frac{1}{2}W, \frac{1}{2}H)$, $RD(-f, \frac{1}{2}W, -\frac{1}{2}H)$, where $W$ and $H$ are the image width and height and $f$ denotes the camera focal length, which can be computed using the pinhole imaging principle [8] as: $f = \frac{W}{2\tan\left(\frac{1}{2}FoV\right)}$. $FoV$ is the camera field of view. Next, the equations of the lines connecting the image center and the four corner points to the optical center $O_c(0,0,0)$ can be obtained in frame $\{c\}$ (light blue dashed lines in Fig. 3(a)):

$$\begin{cases} l_{LUO_c} : \frac{x}{-f} = \frac{2y}{-W} = \frac{2z}{H} \\ l_{LDO_c} : \frac{x}{-f} = \frac{2y}{-W} = \frac{2z}{-H} \\ l_{RUO_c} : \frac{x}{-f} = \frac{2y}{W} = \frac{2z}{H} \\ l_{RDO_c} : \frac{x}{-f} = \frac{2y}{W} = \frac{2z}{-H} \\ l_{CO_c} : y = 0, z = 0 \end{cases} , \tag{4}$$

where $l_{LUO_c}$ is the line connecting $LU$ to $O_c$, similarly for $l_{LDO_c}$, $l_{RUO_c}$, $l_{RDO_c}$ and $l_{CO_c}$. Thus, the projections of the points can be obtained by intersecting the lines with the ground plane.

Therefore, we next derive the expressions for the ground plane and the target. For clarity, we adopt a unified representation in frame $\{c\}$. In DAT scenes, the road surfaces are smooth. Thus, in the world frame $\{w\}$ (see Fig. 3(a)), the ground plane $G_w$ is defined as: $z = h$, where $h$ denotes the ground height. For simplicity, we here express $G_w$ in $\{c\}$ as $G_c$: $A_g x + B_g y + C_g z + D_g = 0$, with $A_g, B_g, C_g, D_g$ derived in Appendix C.2. Furthermore, the target coordinates $P_v = (x_v, y_v, z_v, 1)^T$ in $\{w\}$ can be transformed to $\{c\}$ using *homogeneous transformation matrix* [7] $T_{cw}$: $P_g = T_{cw}^{-1} P_v$.

Subsequently, the ground projections can be obtained by intersecting lines in Eq. 4 and $G_c$:

$$\begin{cases} LU_g : (-f, -\frac{1}{2}W, \frac{1}{2}H)t_{lu}, & t_{lu} = D_g(A_g f + \frac{1}{2}B_g W - \frac{1}{2}C_g H)^{-1} \\ LD_g : (-f, -\frac{1}{2}W, -\frac{1}{2}H)t_{ld}, & t_{ld} = D_g(A_g f + \frac{1}{2}B_g W + \frac{1}{2}C_g H)^{-1} \\ RU_g : (-f, \frac{1}{2}W, \frac{1}{2}H)t_{ru}, & t_{ru} = D_g(A_g f - \frac{1}{2}B_g W - \frac{1}{2}C_g H)^{-1} \\ RD_g : (-f, \frac{1}{2}W, -\frac{1}{2}H)t_{rd}, & t_{rd} = D_g(A_g f - \frac{1}{2}B_g W + \frac{1}{2}C_g H)^{-1} \\ C_g : (-\frac{D_g}{A_g}, 0, 0) \end{cases} , \tag{5}$$

where $LU_g$, $LD_g$, $RU_g$, $RD_g$ and $C_g$ are the projections of $LU$, $LD$, $RU$, $RD$ and $C$. Using target coordinates $P_g$ and Eq. 5, the reward is computed as Eq. 3. See Appendix C.2 for details.

### 4.3 Theoretical Guarantees on Reward Design

Existing methods [39, 19] assume a fixed forward camera view and use distance-based rewards. However, when the view changes, these rewards may fail due to the affine transformation effect in

image projection. We hereby provide a theoretical analysis to show that commonly used distance-based rewards will fail when the camera deviates from a fixed horizontal forward view.

To this end, we define $\mathcal{R}_d(\cdot)$ as a distance-based reward using Euclidean distance between the target and the image center projection. A distance-based reward $\mathcal{R}_d(\cdot)$ satisfying Eq. 2 may still assign higher rewards to targets farther from the center under the metric $\phi(\cdot, \cdot)$, rendering it ineffective. In contrast, any deviation-based reward $R_\phi(\cdot)$ satisfying Eq. 2 can effectively reflect the target position.

**Proposition 1** *The commonly used Euclidean distance $d(\cdot, \cdot)$ between the target and the image center proposition does not align with the deviation $\phi(\cdot, \cdot)$ of the target from the image center projection, when the camera is not at a fixed horizontal forward viewpoint. That is:*

$$\exists P_1, P_2 \in \mathcal{I}_p, \text{ s.t. } \phi_1 < \phi_2, d(P_1, C_g) > d(P_2, C_g), \tag{6}$$

*where $\phi_i = \phi(P_i, C_g)$, $P_i$ are points in the projection region $\mathcal{I}_p$, $C_g$ is the image center projection. See Appendix C.1 for theoretical proof.*

**Remark 1.** A distance-based reward $\mathcal{R}_d(\cdot)$ satisfying Eq. 2 results in targets closer to the center receiving lower rewards, when the camera is not at a fixed horizontal forward viewpoint. That is:

$$\exists P_1, P_2 \in \mathcal{I}_p, \text{ s.t. } \phi_1 < \phi_2, \mathcal{R}_d(d_1) < \mathcal{R}_d(d_2), \tag{7}$$

where $d_i = d(P_i, C_g)$, and $\phi_i = \phi(P_i, C_g)$ . This illustrates the failure of the distance-based reward under these viewpoints. See Appendix C.1 for theoretical proof.

**Qualitative Analysis.** According to the **Theoretical Analysis** above, rewards should decrease monotonically along the deviation contours in Fig. 4(b) as the target moves toward the projection boundary. Thus, the reward contours must align with the deviation contours. The contours of $r_{gc}(\cdot)$ in Fig. 4(b) perfectly align, indicating accurate position feedback. In contrast, D-VAT [19] (see Fig. 4(a)) shows misaligned contours, explaining its failure as noted in **Remark 1**.

## 4.4 Training with Curriculum Learning

DAT scenes contain numerous dynamic targets and obstacles, hindering convergence and performance. Progressively training the agent from simpler to more complex environments enhances performance and accelerates learning for the final task [63]. Therefore, we propose a Curriculum-Based Training (CBT) strategy to optimize reinforcement learning training in complex environments.

To address the challenges, we employ the Proximal Policy Optimization (PPO) [55] algorithm, known for its efficiency in control tasks. To further enhance agent adaptability and robustness, we apply domain randomization during agent training. Specifically, we randomize the drone's initial position and orientation relative to the target to promote diverse behaviors. Additionally, we randomize the gimbal pitch angle to improve the agent's spatial perception. See Appendix C.2 for further details.

Given the scene complexity, we adopt a CBT strategy, which divides the model training into two stages. The first stage consists of a simplified environment with straight line target trajectories and no obstacles. The agent learns to center the target through the reward $r_t$ in Eq. 3. In the second stage, the agent encounters more varied target movements and complex visual information, such as tree occlusions and crosswalks. The goal of the agent is to develop stronger generalization abilities based on task understanding in the first stage. See Algorithm 1 for the pseudocode of the CBT strategy.

# 5 Experiments

## 5.1 Experimental Settings

**Experimental Setup.** We conduct cross-scene and cross-domain tests. The former tests an agent trained under daytime conditions in unseen scenes with the same weather. The latter evaluates the agent in the same scene under varying weather conditions. See Appendix E.1 for details.

**Metrics.** We use cumulative reward ($CR = \sum_{t=1}^{E_l} r_{gc}$) and tracking success rate ($TSR = \frac{1}{E_{ml}} \sum_{t=1}^{E_l} r_{dt} \times 100\%$) to evaluate the agent performance. $CR$ primarily reflects how well the agent centers the target over episode length $E_l$, while $TSR$ measures the ability to keep the target in view, with $r_{dt} = 1$ meaning the target is within the view (See Appendix C), and $E_{ml}$ denoting the

Table 2: Results of within and across scenes on DAT benchmark.

| Method | citystreet CR | citystreet TSR | desert CR | desert TSR | village CR | village TSR | downtown CR | downtown TSR | lake CR | lake TSR | farmland CR | farmland TSR |
|---|---|---|---|---|---|---|---|---|---|---|---|---|
| **Within Scene** | | | | | | | | | | | | |
| AOT | $49_{\pm3}$ | $0.25_{\pm0.02}$ | $9_{\pm1}$ | $0.06_{\pm0.00}$ | $46_{\pm5}$ | $0.23_{\pm0.03}$ | $54_{\pm5}$ | $0.29_{\pm0.01}$ | $47_{\pm3}$ | $0.24_{\pm0.02}$ | $60_{\pm25}$ | $0.23_{\pm0.01}$ |
| D-VAT | $48_{\pm8}$ | $0.26_{\pm0.02}$ | $47_{\pm13}$ | $0.26_{\pm0.04}$ | $44_{\pm8}$ | $0.22_{\pm0.05}$ | $9_{\pm1}$ | $0.06_{\pm0.01}$ | $46_{\pm8}$ | $0.26_{\pm0.06}$ | $13_{\pm1}$ | $0.07_{\pm0.00}$ |
| **Ours** | $\mathbf{279}_{\pm110}$ | $\mathbf{0.80}_{\pm0.30}$ | $\mathbf{307}_{\pm124}$ | $\mathbf{0.84}_{\pm0.29}$ | $\mathbf{239}_{\pm134}$ | $\mathbf{0.73}_{\pm0.32}$ | $\mathbf{203}_{\pm119}$ | $\mathbf{0.65}_{\pm0.30}$ | $\mathbf{181}_{\pm116}$ | $\mathbf{0.61}_{\pm0.31}$ | $\mathbf{243}_{\pm117}$ | $\mathbf{0.68}_{\pm0.32}$ |
| **Cross Scene** | | | | | | | | | | | | |
| AOT | $48_{\pm5}$ | $0.24_{\pm0.02}$ | $9_{\pm0}$ | $0.06_{\pm0.00}$ | $52_{\pm11}$ | $0.25_{\pm0.03}$ | $52_{\pm6}$ | $0.28_{\pm0.03}$ | $48_{\pm5}$ | $0.24_{\pm0.02}$ | $49_{\pm7}$ | $0.24_{\pm0.03}$ |
| D-VAT | $49_{\pm9}$ | $0.26_{\pm0.04}$ | $48_{\pm8}$ | $0.27_{\pm0.03}$ | $50_{\pm14}$ | $0.25_{\pm0.06}$ | $8_{\pm1}$ | $0.05_{\pm0.00}$ | $51_{\pm14}$ | $0.25_{\pm0.06}$ | $14_{\pm1}$ | $0.07_{\pm0.01}$ |
| **Ours** | $\mathbf{144}_{\pm111}$ | $\mathbf{0.52}_{\pm0.29}$ | $\mathbf{229}_{\pm115}$ | $\mathbf{0.67}_{\pm0.27}$ | $\mathbf{156}_{\pm119}$ | $\mathbf{0.55}_{\pm0.31}$ | $\mathbf{201}_{\pm121}$ | $\mathbf{0.64}_{\pm0.30}$ | $\mathbf{163}_{\pm115}$ | $\mathbf{0.51}_{\pm0.29}$ | $\mathbf{162}_{\pm106}$ | $\mathbf{0.54}_{\pm0.26}$ |

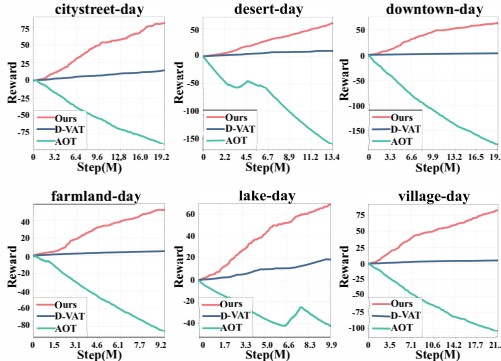

Figure 5: Reward values during training.

Table 3: Results of cross domain on DAT.

| Method | night CR | night TSR | foggy CR | foggy TSR | snow CR | snow TSR |
|---|---|---|---|---|---|---|
| AOT | $42_{\pm4}$ | $0.22_{\pm0.02}$ | $44_{\pm7}$ | $0.22_{\pm0.02}$ | $44_{\pm7}$ | $0.22_{\pm0.02}$ |
| D-VAT | $35_{\pm7}$ | $0.19_{\pm0.03}$ | $37_{\pm7}$ | $0.19_{\pm0.03}$ | $34_{\pm6}$ | $0.20_{\pm0.03}$ |
| **Ours** | $\mathbf{217}_{\pm125}$ | $\mathbf{0.64}_{\pm0.32}$ | $\mathbf{243}_{\pm114}$ | $\mathbf{0.76}_{\pm0.26}$ | $\mathbf{178}_{\pm105}$ | $\mathbf{0.60}_{\pm0.26}$ |

Table 4: Results of ablation experiments on DAT.

| Method | Within-Scene CR | Within-Scene TSR | Cross-Scene CR | Cross-Scene TSR | Cross-Domain CR | Cross-Domain TSR |
|---|---|---|---|---|---|---|
| $R_{\text{D-VAT}}$ | $9_{\pm1}$ | $0.06_{\pm0.00}$ | $8_{\pm1}$ | $0.05_{\pm0.00}$ | $9_{\pm0}$ | $0.06_{\pm0.00}$ |
| w/o CBT | $46_{\pm2}$ | $0.23_{\pm0.01}$ | $53_{\pm16}$ | $0.26_{\pm0.07}$ | $46_{\pm2}$ | $0.23_{\pm0.01}$ |
| w/o AR | $106_{\pm88}$ | $0.44_{\pm0.23}$ | $92_{\pm72}$ | $0.37_{\pm0.19}$ | $80_{\pm63}$ | $0.36_{\pm0.19}$ |
| w/o HR | $174_{\pm118}$ | $0.49_{\pm0.30}$ | $148_{\pm129}$ | $0.48_{\pm0.32}$ | $184_{\pm124}$ | $0.57_{\pm0.30}$ |
| w/o VR | $211_{\pm138}$ | $0.63_{\pm0.35}$ | $161_{\pm115}$ | $0.54_{\pm0.32}$ | $203_{\pm117}$ | $0.60_{\pm0.32}$ |
| w/o PR | $139_{\pm119}$ | $0.61_{\pm0.33}$ | $124_{\pm85}$ | $0.48_{\pm0.25}$ | $145_{\pm122}$ | $0.52_{\pm0.28}$ |
| **Ours** | $\mathbf{243}_{\pm117}$ | $\mathbf{0.68}_{\pm0.32}$ | $\mathbf{162}_{\pm106}$ | $\mathbf{0.54}_{\pm0.26}$ | $\mathbf{222}_{\pm110}$ | $\mathbf{0.65}_{\pm0.27}$ |

maximum episode length. Agents are initialized at four relative angles to the target ($[0, \frac{\pi}{2}, \pi, \frac{3\pi}{2}]$ rad), with 10 episodes per angle (40 total). The mean and variance of these results are calculated for each map, and the final cross-scene and cross-domain performance are averaged across different scenes.

**Baselines.** We reproduce two SOTA methods: AOT [39] and D-VAT [19]. Both baselines and other methods [81, 18] use distance-based rewards. As concluded in Section 4.3, they may fail in tilted top-down views. Thus, these baselines sufficiently highlight GC-VAT superiority. See Appendix D.

## 5.2 Comparison Experiments

We compare our GC-VAT with the SOTA methods for within-scene performance and cross-scene cross-domain generalization performance on DAT benchmark. As shown in Fig. 5, our method achieves consistently higher and steadily increasing rewards throughout training, demonstrating its effectiveness. Both AOT and D-VAT methods fail to learn effective policies due to the misleading feedback from their distance-based rewards. In particular, AOT learns to quickly drive the target out of view, resulting in a rapidly declining reward curve. The results validate the theoretical analysis in Section 4.3. It is worth noting that although AOT and D-VAT exhibit low variance in their experimental results, consistently low rewards typically indicate a failure to learn effective tracking policies.

**Within-scene performance.** We train the model on all scenes and evaluate it on the original scene. Our GC-VAT performs significantly better than other methods as shown in Table 2. For the $CR$, the average performance improvement on six maps relative to the D-VAT method is $591\%(35\rightarrow242)$. Regarding the $TSR$, the average enhancement is $279\%(0.19\rightarrow0.72)$.

**Cross-scene performance.** Our method demonstrates strong cross-scene generalization, as shown in Table 2. Specifically, GC-VAT achieves a $376\%(37\rightarrow176)$ improvement in average $CR$ and a $200\%(0.19\rightarrow0.57)$ improvement in average $TSR$ compared to D-VAT.

**Cross-domain performance.** As shown in Table 3, our method outperforms existing methods significantly in cross-domain generalization. Specifically, GC-VAT demonstrates an average $CR$ enhancement of $509\%(35\rightarrow213)$ relative to D-VAT and $TSR$ boost of $253\%(0.19\rightarrow0.67)$.

Table 5: Performance under wind disturbances and target distractors.

| | CR | TSR |
|---|---|---|
| w/ Forward | $302_{\pm 94}$ | $0.91_{\pm 0.18}$ |
| w/ Lateral | $304_{\pm 82}$ | $0.91_{\pm 0.19}$ |
| w/ Yaw | $301_{\pm 120}$ | $0.88_{\pm 0.23}$ |
| w/ Distractor | $293_{\pm 120}$ | $0.91_{\pm 0.15}$ |
| **Ours** | $\mathbf{316}_{\pm 84}$ | $\mathbf{0.94}_{\pm 0.14}$ |

Table 6: Performance under rainy conditions and unseen targets. We evaluate the model trained on citystreet-day.

| Method | Within-Scene | | Cross-Scene | | Cross-Domain | |
|---|---|---|---|---|---|---|
| | CR | TSR | CR | TSR | CR | TSR |
| w/ rain | $266_{\pm 110}$ | $0.74_{\pm 0.29}$ | $139_{\pm 109}$ | $0.45_{\pm 0.30}$ | $274_{\pm 103}$ | $0.77_{\pm 0.29}$ |
| Unseen Target | $222_{\pm 92}$ | $0.79_{\pm 0.25}$ | $131_{\pm 89}$ | $0.50_{\pm 0.33}$ | $207_{\pm 94}$ | $0.79_{\pm 0.27}$ |
| **Ours** | $\mathbf{279}_{\pm 110}$ | $\mathbf{0.80}_{\pm 0.30}$ | $\mathbf{144}_{\pm 111}$ | $\mathbf{0.52}_{\pm 0.29}$ | $\mathbf{258}_{\pm 110}$ | $\mathbf{0.82}_{\pm 0.23}$ |

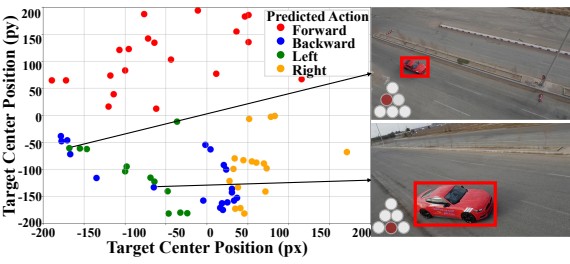

Figure 6: Results on real-world images.

Table 7: Effectiveness of GC-VAT on Sim2Real test. We select eight video sequences from each dataset for evaluation.

| Video | VOT [30] | DTB70 [35] | UAVDT [20] |
|---|---|---|---|
| | **Average Correct Action Rate** | | |
| Random | 0.413 | 0.426 | 0.421 |
| **Ours** | **0.795** | **0.833** | **0.802** |

## 5.3 Ablation Experiments

We conduct ablation experiments on Goal-Centered Reward to validate the results of the analysis presented in Section 4.3. Moreover, we verify whether the Curriculum-Based Training strategy and domain rondomization from Section 4.4 lead to a significant performance improvement. We present results on the *farmland* map in Table 4, with additional results provided in Appendix E.3.

**Effectiveness of reward design.** We contrast the performance of GC-VAT method when using the reward defined in Eq. 3 and that in [19]. As shown in Table 4, significant performance enhancements (about 800% improvement in $TSR$ across-scene and cross-domain) are evident with the utilization of Eq. 3. These results strongly corroborate the analysis in Section 4.3 and underscore the effectiveness of the proposed reward. See Appendix E.3 for more experimental results.

**Effectiveness of CBT strategy and domain randomization.** As shown in Table 4, without the CBT strategy, the model fails to learn effective tracking policies, resulting in consistently low rewards across different tests. In addition, our domain randomization approach yields significant improvements. Specifically, $AR$, $HR$, $VR$, and $PR$ denote the randomization of the drone's initial angle, horizontal and vertical distance relative to the target, and gimbal pitch angle, respectively. Among these, $AR$ contributes the most to performance gains, indicating that encouraging diverse actions through angle randomization facilitates the agent's exploration of optimal policies.

**Robustness under wind gusts and precipitation.** To further investigate the impact of real-world disturbances on the GC-VAT method, we conduct rigorous tests under wind gusts and sensor degradation caused by precipitation. Specifically, we simulate wind effects by applying randomized perturbations along the forward, lateral, and yaw directions during testing. The results are summarized in Table 5, where the model is trained on citystreet-day and evaluated on citystreet-foggy with added wind perturbations. The Tracking Success Rate (TSR) drops by less than 0.06, demonstrating that GC-VAT maintains strong robustness under significant wind disturbances. See Appendix E.3 for more details.

To simulate the blurring caused by raindrops, we follow established practices in test-time adaptation literature [34]. Specifically, we train the policy on citystreet-day map and evaluate under synthetically generated rain in within-scene, cross-scene, and cross-domain settings. To ensure realism, we exclude snowy conditions from the cross-domain evaluation, as snow and rain rarely co-occur in real-world environments. The results in Table 6 show only marginal performance degradation (less than 0.07 in TSR) under rain simulation, confirming that GC-VAT is robust to blurring caused by raindrops.

**Robustness to distractors and novel targets.** As shown in Table 5, our model maintains high tracking performance even when a similar-looking vehicle is introduced near the target, demonstrating its ability to effectively distinguish the true target from confusers. In addition, we evaluate GC-VAT on an unseen target class (bus). As shown in Table 6, our model maintains strong tracking performance, with a TSR drop of less than 0.03 when encountering this novel object.

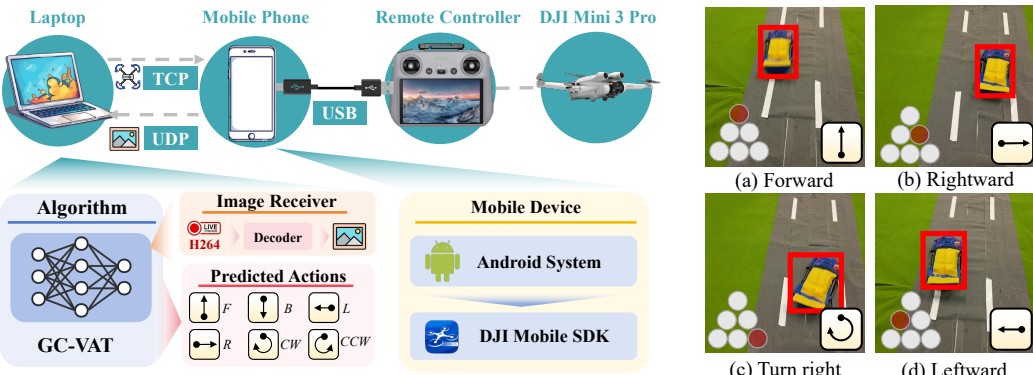

Figure 7: Schematic of the real-world deployment pipeline.

Figure 8: Results on real drones.

## 5.4 Experiments in Real-world Scenarios

**Effectiveness on real-world images.** Due to the difficulty real-robot evaluation, we follow [39] and validate GC-VAT on real images. We perform zero-shot transfer tests using 8 videos each from VOT [30], DTB70 [35] and UAVDT [20] datasets. Although camera control is unavailable in recorded videos, we can feed frames into the model and verify the reasonableness of its predicted actions.

The output actions for the VOT video *car16* are shown in Fig. 6. Each point represents the target position in the image, with colors indicating different actions. As Fig. 6 illustrates, when the target is located on the left (right) side, the tracker tends to move left (right), attempting to bring the target to the center. Quantitatively, we use the *Correct Action Rate*, i.e., the accuracy of predicted actions, to evaluate the performance. As shown in Table 7, GC-VAT achieves an average *Correct Action Rate (CAR)* of $81.0\%$ across 24 videos, demonstrating its effectiveness. More importantly, it is significantly superior to random policy ($p < 0.001$) as verified by a t-test. See Appendix E.4 for more results.

**Effectiveness on real drones.** Furthermore, as a critical step beyond image-based evaluation, we conduct real-world experiments on a *DJI Mini 3 Pro* [13] drone. As shown in Fig. 7, we deploy GC-VAT on a laptop equipped with an RTX 3050 GPU and an Intel i5 CPU, use the DJI Mobile SDK [12] to obtain images, and control the drone with the predicted actions. The entire pipeline operates at over 30 FPS. As Fig. 8 illustrates, the model can output actions that maintain the target at the image center. Quantitatively, GC-VAT achieves an average zero-shot TSR of $88.4\%$ and a CAR of $81.3\%$. This successful zero-shot Sim-to-Real transfer validates the practical applicability of our approach.

## 6 Conclusion and Potential Impacts

In this paper, we propose the first open-world drone active air-to-ground tracking benchmark, called DAT. DAT benchmark encompasses 24 city-scale scenes, featuring targets with human-like behaviors and high-fidelity dynamics simulation. DAT also provides a digital twin tool for unlimited scene generation. DAT benchmark has the potential to impact several key areas, including: 1) Forgetting in Reinforcement Learning, 2) Robustness in Reinforcement Learning, 3) Multi-Agent Reinforcement Learning, and 4) Sim-to-Real Deployment. Additionally, we propose a reinforcement learning-based drone tracking method called GC-VAT, aiming to improve the performance of drone tracking targets in complex scenarios. Specifically, we design a Goal-Centered Reward to provide precise feedback across viewpoints to the agent, enabling it to expand perception range through unrestricted perspectives. Then we propose qualitative and theoretical methods to analyze the reward effectiveness. Moreover, inspired by curriculum learning, we implement a Curriculum-Based Training strategy that progressively improves agent performance in increasingly complex scenarios. Experiments on the simulator and real-world images validate the analysis and demonstrate that our method is significantly superior to the SOTA methods.

## Acknowledgements

This work was partially supported by the Joint Funds of the National Natural Science Foundation of China (Grant No.U24A20327).

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

# Supplementary Materials for
# "Open-World Drone Active Tracking with
# Goal-Centered Rewards"

## Contents

## A  Related Work

### A.1  Passive Object Tracking

Most of the proposed visual tracking benchmarks belong to passive visual tracking. LaSOT [23] and OTB2015 [71] benchmarks contain a large number of ground-based videos. These benchmarks include target videos, and the tracking algorithms utilize both the video frames and the target labels for tracking. However, ground cameras tend to be affected by occlusion and suffer from the shortcoming of limited perceptual range, so the need for drone viewpoint tracking is gradually increasing in practical applications. UAV123 [43] and VisDrone2019 [21] benchmarks are proposed for drone viewpoint, expanding the spatial dimension of perception. Meanwhile, the single-object tracking benchmarks have difficulties for many targets. MOT20 [17] and TAO [16] benchmarks are proposed for multi-object tracking to solve the above problems. In addition, the above benchmarks include videos from the RGB camera. The RGB camera's recognition capabilities are limited in complex scenes, such as ocean environments, and challenging weather conditions, including nighttime and foggy. IPATCH [47] provides extra infrared images and other sensors like GPS to supplement the information of the sea scene. Huang et al. propose Anti-UAV410 [28], which provides infrared camera images for drone tracking.

Visual object tracking methods can be categorized into three main types: Tracking by Detection, Detection and Tracking (D&T), and pure tracking. Tracking by Detection methods [5, 69, 6] treat tracking as a sequence of independent detection tasks. These methods use object detection algorithms [50, 52] to identify the target object in each frame, connecting the detections through data association methods [31, 70] for continuous tracking. While effective in multi-target tracking, these methods

may suffer from high computational demands and issues with target occlusion. D&T approaches [66, 79, 48] integrate detection and tracking, creating end-to-end models that ensure seamless information flow and reduce redundant calculations through shared feature extraction networks. Pure tracking methods can be categorized into two main types: Correlation Filters (CF) [72, 44, 27] and Siamese Networks (SN) [32, 4, 62]. CF-based models train correlation filters on regions of interest, while SN-based models compare target templates with search areas to enable precise single-target tracking.

## A.2 Visual Active Tracking

Passive visual tracking often falls short in real-world scenarios due to the highly dynamic nature of most targets. Visual Active Tracking (VAT) aims to autonomously follow a target object by controlling the motion system of the tracker based on visual observations [40, 80, 75]. Thus, VAT offers a more practical yet challenging solution for effective tracking in dynamic environments. Maalouf et al. [40] propose a two-stage tracking method (named FAn), which is based on a tracking model and a PID control model. This method accomplishes the fusion of perception and decision-making by transferring control information from the visual tracking model to the control model. However, the visual network necessitates extensive human labeling effort and the control model requires parameter adjustments for each scene, significantly constraining the model's generalizability. Recently, many approaches [39, 18, 80, 19] model the VAT task as a Markov Decision Process and employ end-to-end training with reinforcement learning, resulting in a significant enhancement of the agent's generalizability.

The complexity and diversity of VAT benchmarks are crucial for training agents with high generalizability. One common approach [19, 18, 80] to enhancing environmental diversity involves modifying texture features and lighting conditions within a single scene. However, these methods often result in low scene fidelity and unrealistic object placement. While UE4 [24] is used to create photorealistic environments in some benchmarks [80, 39], these benchmarks still face limitations in diversity and map size. Furthermore, the scenarios provided by these methods are often task-specific, offering limited configurability and lacking a unified benchmark for VAT tasks.

Existing approaches to VAT frequently neglect the randomness of target trajectories and the scalability of platforms. Target trajectories are typically predefined by rule-based patterns [19, 18, 39], which significantly restrict the exploration space. Zhong et al. [80] introduce learnable agents as targets, increasing trajectory randomness but adding additional cost. Most benchmarks provide only a single category of target [19, 18, 80, 39], limiting scalability and necessitating repetitive work for environment development. Zhou et al. [82] utilize CoppeliaSim [2] to provide five categories of noncooperative space objects. However, the use of a solid black background makes it unsuitable for general VAT scenarios. In contrast, our environment supports diverse, real-world target types and offers unified, lightweight management of target behaviors, ensuring both rationality and randomness in their actions.

## A.3 Reinforcement learning in Visual Tracking

Reinforcement learning (RL) is widely used in large language models [26] and robot control [53] to improve exploration performance. It is also commonly applied in visual object tracking [76, 51, 78]. Song et al. [59] propose a decision-making mechanism based on hierarchical reinforcement learning (HRL), which achieves state-of-the-art performance while maintaining a balance between accuracy and computational efficiency. However, the actions generated by reinforcement learning in the aforementioned work cannot directly influence the camera's viewpoint, thereby failing to fully leverage the decision-making capabilities. Real-world applications increasingly require robust tracking in highly dynamic scenes, motivating researchers to explore reinforcement learning agents for effectively synchronizing visual perception and decision-making in VAT tasks. Dionigi et al. [19] demonstrate the feasibility of reinforcement learning for drone VAT missions. However, the assumption of a fixed-forward perspective limits its applicability in real-world tasks.

## A.4 Curriculum Learning in Robot Control

Curriculum Learning (CL) is a training strategy that mimics a human curriculum by training models on simpler subsets of data at first and gradually expanding to larger and more difficult subsets of data

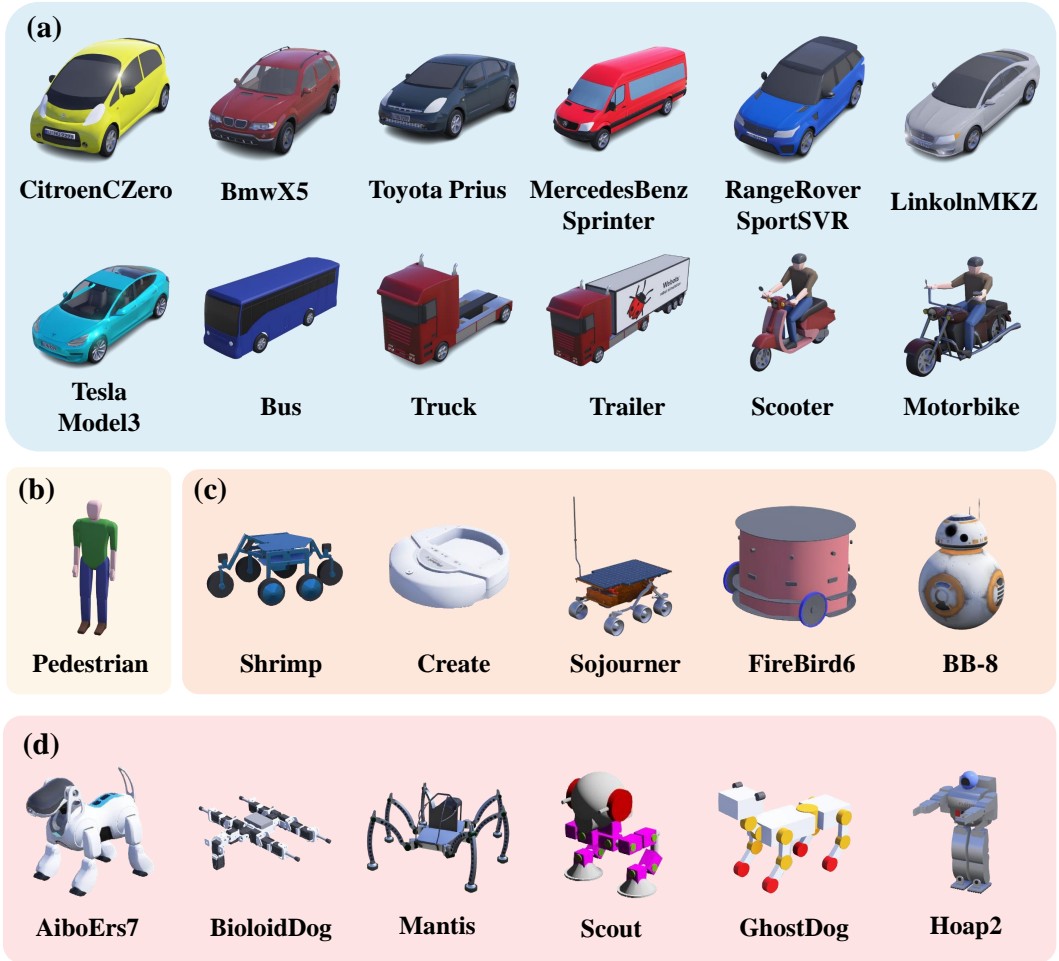

Figure 9: Examples of DAT benchmark targets. (a) Illustration of tracking targets for 10 types of *automobile* and 2 types of *motorbike*. (b) Illustration of tracking targets for the *pedestrian* type. (c) Illustration of tracking targets for 5 types of *wheeled robot*. (d) Illustration of tracking targets for 6 types of *legged robot*.

until they are trained on the entire dataset. CL is widely used in large language models [64] and robot control. As for robot control, reinforcement learning training is difficult due to the complexity of the training scenarios and the large action spaces. Therefore, curriculum learning is often required to reduce the difficulty of agent training. For instance, many works improve the walking ability of legged robots by adjusting terrain parameters through curriculum learning [53, 41]. Other studies improve the pushing and grasping performance of robotic arms by progressively increasing task difficulty [38, 61, 45].

In this paper, Curriculum Learning is introduced in the VAT task, and the training environment is transitioned from simple features to complex scenarios to achieve successful tracking of agent in complex outdoor environments.

## B   More Details of DAT Benchmark

**More details of the digital twin tool.** Our digital twin tool is based on the *osm_importer* tool in the *webots* simulation software. Users first need to download the map description file (*.osm* file) for a specific area from the *OpenStreetMap* website. Then, the tool preprocesses the map according to the configuration, modifying information such as the number of lanes and lane directions, and converts the processed file into a road network file (*.net.xml* file) that can be read by SUMO. Following this, the tool adds traffic lights and intersection traffic rules to the road network based on the configuration,

Table 8: State parameters of DAT benchmark.

| Category | Sensor | Parameter | Type | Description | Potential Tasks |
|---|---|---|---|---|---|
| Vision | Camera | Image | Mat | Images | Default sensor |
| | LiDAR | LidarCloud | vector2000 | Point cloud (m) | Obstacle avoidance |
| Motion | GPS | Position | vector3 | Position (m) | Visual navigation |
| | | Linear | vector3 | Linear velocity (m/s) | Visual navigation |
| | Accelerometer | Acc | vector3 | Acceleration (m/s$^2$) | Visual navigation |
| | Gyroscope | Angular | vector3 | Angular velocity (rad/s) | Posture stabilization |
| | IMU | Angle | vector3 | Euler angles (rad) | Posture stabilization |
| | | Orientation | vector4 | Quaternion representation | Posture stabilization |

Table 9: Reward parameters of DAT benchmark. The homogeneous transformation matrices (HTM) $T_{cw}$ and $T_{tw}$ are $4\times4$ square matrices. Therefore, their data type double[16] corresponds to a double array of length 16.

| Parameter | Type | Description |
|---|---|---|
| cameraWidth | double | image width($px$) |
| cameraHeight | double | image height($px$) |
| cameraFov | double | camera field of view($rad$) |
| cameraF | double | estimated camera focal length($px$) |
| $T_{cw}$ | double[16] | HTM of the camera relative to the world frame |
| $T_{tw}$ | double[16] | HTM of the vehicle relative to the world frame |
| cameraMidGlobalPos | vector3d | ground projection of camera center mapped in the world frame |
| carMidGlobalPos | vector3d | coordinates of the vehicle center in the world frame |
| cameraMidPos | vector3d | coordinates of the camera center in the world frame |
| carDronePosOri | vector4d | 1D orientation + 3D position of vehicle in the drone frame |
| crash | double | whether tracker collides with a building |
| carDir | double | car direction(*0-stop,1-go straight,2-turn left,3-turn right*) |
| carTypename | string | tracking target type |

ensuring that the traffic flow operates correctly when the map is converted into a 3D scene. Finally, the tool reads the road, vegetation, and building information and converts them into *PROTO* assets for webots, which can then be correctly recognized and used by the DAT benchmark.

**Scenario Construction.** Among the DAT scenes, three scenarios: *citystreet*, *downtown*, and *lake* are directly derived from real-world locations with the digital twins tool. Specifically, the *citystreet* scenario is based on a small town in Los Angeles, the *downtown* scenario is derived from Manhattan, and the *lake* scenario is modeled after Wolf Lake Memorial Park in Indiana. In contrast, the *village*, *desert*, and *farmland* maps possess complex and unique features that are not adequately captured by OpenStreetMap (OSM) data. For example, the *village* map features mountainous terrain with tunnels, while the *farmland* map is characterized by diverse multicolored patterns. To overcome these limitations, we use Creo software [29] to model detailed scene elements, which are then integrated into the webots for constructing realistic maps.

**Targets.** All tracking target illustrations are presented in Fig. 9. Specifically, Fig. 9(a) presents *automobile* and *motorbike* tracking targets, including passenger vehicles (the first seven cars), buses, trucks, trailers, and motorcycles (such as scooters and motorbikes). These two categories of tracking targets leverage Simulation of Urban Mobility (SUMO) [36] for road behavior modeling and inter-action management with other targets. In contrast, Fig. 9(b)-(d) display *pedestrian*, *wheeled robot*, and *legged robot* tracking targets, respectively. These three types of targets utilize SUMO paths for position initialization and rely on specific controllers for action and behavior management.

**Sensors.** In the real world VAT tasks, a single camera cannot ensure the agent's stability and robustness. Thus, integration with other sensors is often required. The DAT benchmark provides common sensors that can obtain the drone's state parameters relative to the world coordinate system. The drone's position and velocity are determined using GPS, while its acceleration is measured by an accelerometer, providing essential self-referential data for visual navigation tasks. Angular velocity is recorded via a gyroscope, and Euler angles obtained from the IMU are converted into quaternions to facilitate state estimation and ensure orientation stability. Additionally, the *RPLIDAR A2*, provided by DAT, generates point cloud data, which supports tasks such as obstacle avoidance and navigation

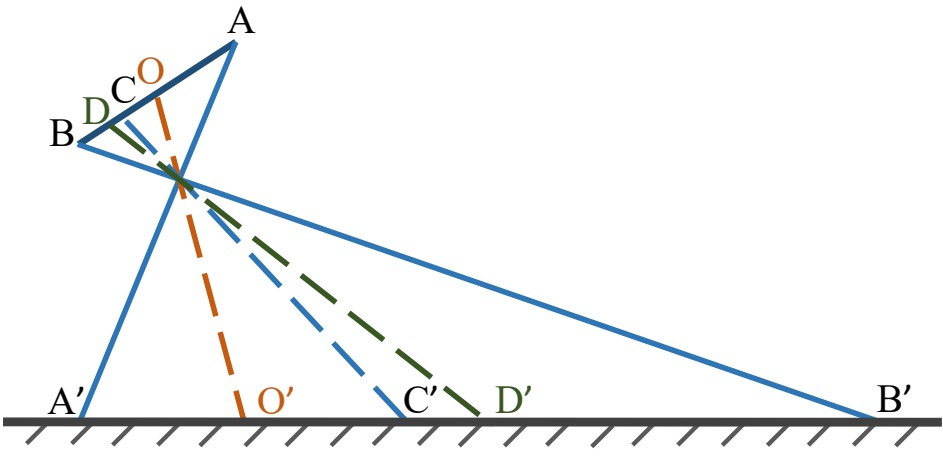

Figure 10: Diagram for the theoretical proof of **Proposition 1**.

by delivering detailed environmental information. The specific sensors, parameters and potential tasks are in Table 8.

**Additional Parameters.** The training process of VAT agents often requires additional parameters for effective reward design. To facilitate this, DAT benchmark provides 4 categories comprising a total of 13 parameters, supporting diverse reward design strategies, as detailed in Table 9.

First are the camera parameters, which mainly include image width `cameraWidth`, image height `cameraHeight`, field of view `cameraFov`, and focal length `cameraF`. Utilizing these, the camera plane can be projected onto the ground to aid in reward construction.

Next is the homogeneous transformation matrix (HTM). In the reward design, coordinate transformations are often required to express physical quantities within a unified coordinate system, enabling consistent calculations. For example, prior studies [19, 39, 18] transform the position, velocity, and acceleration of targets into the tracker's coordinate system to construct rewards. To support such operations, DAT benchmark provides $T_{cw}$, the HTM mapping the drone camera coordinate system to the world coordinate system, and $T_{tw}$, the HTM mapping the tracking target's coordinate system to the world coordinate system.

Additionally, for the state of the tracker itself, `cameraMidPos` represents the position of the drone camera's optical center in the world coordinate system. The parameter `crash` indicates whether the drone collides with any buildings in the scene, which can be used in reward design for obstacle avoidance tasks.

Lastly, for ease of model training in simulations, reward design often depends on some privileged information, i.e., variables that are almost impossible to obtain in real-world settings. Thus, DAT benchmark also provides such adaptations. For example, `carMidGlobalPos` gives the target's position in the world coordinate system, and `carDronePosOri` represents the target's orientation and position relative to the drone coordinate system, frequently used in VAT reward design [19, 39, 18]. Furthermore, information on the target's direction and type is provided.

**Task Configuration.** We encapsulate the scenes, tasks, and domain randomization into Python classes, and provide 3 different environment classes for different algorithm requirements. The base environment class directly interacts with webots and is designed to support asynchronous reinforcement learning algorithms, such as the asynchronous advantage actor-critic (A3C) algorithm [42]. The Gymnasium environment class wraps the base environment class into a Gymnasium [60] interface, enabling direct compatibility with popular reinforcement learning libraries, such as Stable-Baselines3 [49] and Tianshou [68] for efficient algorithm development and evaluation. The parallel environment class encapsulates the base environment class to enable parallel execution, providing direct support for synchronous algorithms, such as proximal policy optimization (PPO) [55] and soft actor-critic (SAC) [25]. Additionally, the scenario selection, tracker and target configuration, SUMO parameters, task additional parameters, and randomization methods can all be efficiently customized through a JSON configuration file.

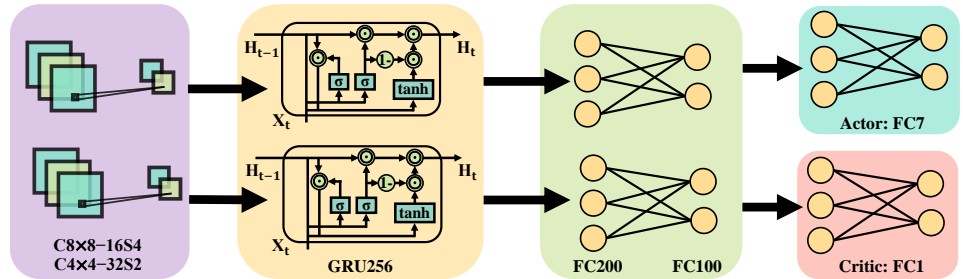

Figure 11: Network structure of Drone Agent.

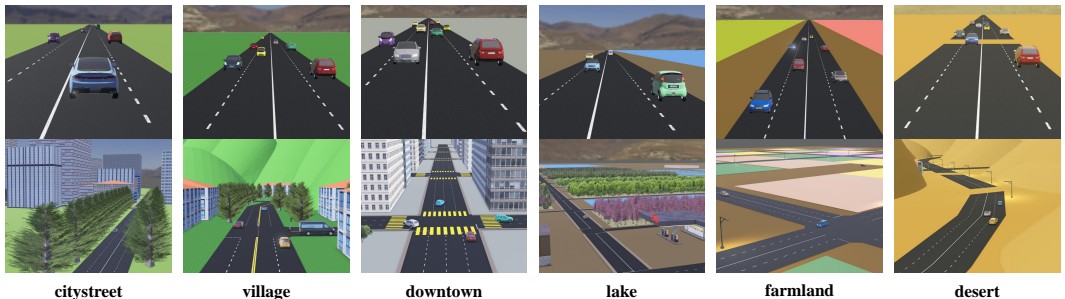

| citystreet | village | downtown | lake | farmland | desert |

Figure 12: Schematic diagram of the training environments for the two-stage of Curriculum Learning

## C More Details of Proposed GC-VAT

### C.1 Theoretical Proof of Reward Design

**Theoretical proof of Proposition 1.** Consider two points $A$ and $B$ on the image symmetry axis (in Fig. 10), which are symmetric with respect to the image center $O$. The projections of these points onto the ground are denoted as $A'$, $B'$ and $O'$, respectively. Take a point $C'$ on line segment $O'B'$ such that the Euclidean distance $d(O', C') = d(A', O')$.

Given:

1. In the image plane, the deviation $\phi(\cdot, \cdot)$ of point $A$ and $B$ from the image center $O$ is the same, i.e. $\phi(A, O) = \phi(B, O)$.

2. In the projection plane, the Euclidean distance from $A'$ and $C'$ to the ideal position $O'$ are equal, i.e. $d(A', O') = d(O', C')$.

It is evident that:

1. For any point $D'$ on line segment $B'C'$, the following relationship holds: $d(O', D') > d(A', O')$.

2. The corresponding point $D$ in the image lies on line segment $BC$, and thus $\phi(D, O) < \phi(A, O)$.

Thus, it is clear that the actual distance between the target and the ideal position is inconsistent with the deviation of the target from the image center in the image.

**Theoretical proof of Remark 1.** According to **Proposition 1**, the following relationship between the Euclidean distance and the deviation holds:

$$\exists P_1, P_2 \in \mathcal{I}_p, \text{s.t. } \phi_1 < \phi_2, d_1 > d_2, \tag{8}$$

where $\phi_i = \phi(P_i, C_g)$ and $d_i = d(P_i, C_g)$. Therefore, for a distance-based reward function $\mathcal{R}_d(\cdot)$ that satisfies the **Reward Design Principle**, it follows that:

$$\exists P_1, P_2 \in \mathcal{I}_p, \text{s.t. } \phi_1 < \phi_2, \mathcal{R}_d(d_1) < \mathcal{R}_d(d_2). \tag{9}$$

Table 10: Total training steps on different scenes. During the training process, we employ a parallel training approach involving 35 agents. Consequently, the reported total training steps represent the cumulative steps taken by all agents combined.

| Scene | citystreet | desert | village | downtown | lake | farmland |
|---|---|---|---|---|---|---|
| **Steps (M)** | 19.2 | 13.4 | 21.3 | 19.8 | 9.9 | 9.2 |

Table 11: Transition steps across different scenes.

| Scene | citystreet | desert | village | downtown | lake | farmland |
|---|---|---|---|---|---|---|
| **T (M)** | 10.0 | 6.2 | 8.0 | 10.3 | 5.6 | 4.1 |

## C.2 More Details

**Network Structure.** The structure of the GC-VAT is shown in Fig. 11. In this figure, C8$\times$8-16$S$4 represents 16 convolutional filters of size 8$\times$8 and stride 4. GRU256 denotes a GRU network with 256 hidden units, and FC200 represents a fully connected layer with 200 neurons.

**Domain Randomization.** While simpler settings facilitate the agent's learning of task objectives, they also heighten the risk of the agent rapidly converging to a suboptimal action distribution, undermining the exploration process. Consequently, implementing domain randomization is essential. This is achieved through the randomization of the drone's initial position and orientation relative to the target, necessitating a broader range of actions to maximize rewards. Moreover, to enhance the agent's spatial perception ability, randomization is also introduced in its gimbal pitch angle.

In our two-stage curriculum learning process, we employ identical domain randomization. The flight altitude is selected from the interval $[13, 22]$m, and the camera pitch angle is chosen from $[0.6, 1.38]$rad. These parameters are consistent throughout each episode. Meanwhile, the drone's initial orientation relative to the target fluctuates within the range $[-\pi, \pi]$rad, and the target's initial position is set between $[-4.5, -2.5] \cup [2.5, 4.5]$m.

**Details on coordinate transformations.** Given two planes $P_0 : \hat{n_0}\mathbf{x}^T + D_0 = 0$ and $P_1 : \hat{n_1}\mathbf{x}^T + D_1 = 0$, along with the HTM $T_{01}$ from $P_0$ to $P_1$. The $T_{01}$ is defined as:

$$T_{01} = \begin{bmatrix} R_{01} & t_{01} \\ 0 & 1 \end{bmatrix}. \tag{10}$$

Hence, the expression of plane $P_1$ can be obtained using the analytical expression of plane $P_0$ and $T_{01}$ as follows:

$$\begin{aligned} \hat{n_1}^T &= R_{01}\hat{n_0}^T, \\ D_1 &= D_0 - \hat{n_1}t_{01}. \end{aligned} \tag{11}$$

Considering the ground plane $G_w : z = h$ in the world coordinate system $\{w\}$, with representation in the camera coordinate system $\{c\}$ denoted as $G_c : A_g x + B_g y + C_g z + D_g = 0$, the vectors of these two planes are $P_{G_w} = (0, 0, 1, -h)$ and $P_{G_c} = (A_g, B_g, C_g, D_g)$.

Furthermore, from Table 9, we can obtain the HTM $\text{T}_{cw}$ from $\{c\}$ to $\{w\}$ defined as follows:

$$T_{cw} = \begin{bmatrix} R_{cw} & t_{cw} \\ 0 & 1 \end{bmatrix}, \tag{12}$$

where $R_{cw}$ is the rotation matrix from $\{c\}$ to $\{w\}$, which can be expressed in row vector form as: $R_{cw} = [r_1, r_2, r_3]^T$. Therefore, the homogeneous transformation matrix (HTM) $T_{wc}$, which represents the transformation from the world coordinate system $\{w\}$ to the camera coordinate system $\{c\}$, can be expressed as follows:

$$T_{wc} = \begin{bmatrix} R_{cw}^T & -R_{cw}^T t_{cw} \\ 0 & 1 \end{bmatrix}. \tag{13}$$

Using Eq. 11 and the matrix $T_{wc}$, the plane $G_c$ can be formulated as $P_{G_c} = (r_3^T, -h + r_3^T R_{cw}^T t_{cw})$.

**Privileged knowledge available for Drone Agent.** During training in the simulator, the drone agent has access to additional information (e.g., the precise location of the target). However, during testing and real-world deployment, such privileged knowledge is **not** available.

Table 12: The detailed results of comparison experiments on $CR$ metric.

| | Within / Cross Scene | | | | | | Cross Domain | | |
|---|---|---|---|---|---|---|---|---|---|
| Train: citystreet | citystreet | desert | village | downtown | lake | farmland | night | foggy | snow |
| AOT | $49_{\pm3}$ | $49_{\pm9}$ | $45_{\pm5}$ | $49_{\pm3}$ | $48_{\pm3}$ | $48_{\pm3}$ | $49_{\pm4}$ | $49_{\pm3}$ | $49_{\pm3}$ |
| D-VAT | $48_{\pm8}$ | $46_{\pm12}$ | $46_{\pm10}$ | $57_{\pm11}$ | $50_{\pm8}$ | $46_{\pm3}$ | $48_{\pm9}$ | $54_{\pm10}$ | $53_{\pm10}$ |
| **GC-VAT** | $\mathbf{279}_{\pm110}$ | $\mathbf{129}_{\pm112}$ | $\mathbf{153}_{\pm119}$ | $\mathbf{135}_{\pm109}$ | $\mathbf{112}_{\pm92}$ | $\mathbf{191}_{\pm122}$ | $\mathbf{257}_{\pm126}$ | $\mathbf{316}_{\pm84}$ | $\mathbf{202}_{\pm119}$ |
| Train: desert | citystreet | desert | village | downtown | lake | farmland | night | foggy | snow |
| AOT | $9_{\pm0}$ | $9_{\pm1}$ | $9_{\pm1}$ | $9_{\pm1}$ | $9_{\pm0}$ | $9_{\pm0}$ | $9_{\pm1}$ | $9_{\pm1}$ | $9_{\pm1}$ |
| D-VAT | $51_{\pm10}$ | $47_{\pm13}$ | $46_{\pm10}$ | $56_{\pm11}$ | $39_{\pm8}$ | $47_{\pm3}$ | $48_{\pm13}$ | $48_{\pm13}$ | $39_{\pm10}$ |
| **GC-VAT** | $\mathbf{278}_{\pm111}$ | $\mathbf{307}_{\pm124}$ | $\mathbf{305}_{\pm94}$ | $\mathbf{119}_{\pm110}$ | $\mathbf{170}_{\pm139}$ | $\mathbf{275}_{\pm121}$ | $\mathbf{182}_{\pm131}$ | $\mathbf{307}_{\pm124}$ | $\mathbf{307}_{\pm97}$ |
| Train: village | citystreet | desert | village | downtown | lake | farmland | night | foggy | snow |
| AOT | $51_{\pm7}$ | $51_{\pm11}$ | $46_{\pm5}$ | $49_{\pm4}$ | $52_{\pm11}$ | $57_{\pm24}$ | $47_{\pm5}$ | $47_{\pm5}$ | $47_{\pm5}$ |
| D-VAT | $46_{\pm8}$ | $45_{\pm9}$ | $44_{\pm8}$ | $69_{\pm42}$ | $45_{\pm8}$ | $45_{\pm3}$ | $44_{\pm8}$ | $44_{\pm8}$ | $43_{\pm8}$ |
| **GC-VAT** | $\mathbf{234}_{\pm122}$ | $\mathbf{160}_{\pm139}$ | $\mathbf{239}_{\pm134}$ | $\mathbf{93}_{\pm102}$ | $\mathbf{153}_{\pm115}$ | $\mathbf{140}_{\pm118}$ | $\mathbf{257}_{\pm122}$ | $\mathbf{257}_{\pm120}$ | $\mathbf{114}_{\pm115}$ |
| Train: downtown | citystreet | desert | village | downtown | lake | farmland | night | foggy | snow |
| AOT | $52_{\pm3}$ | $52_{\pm9}$ | $48_{\pm7}$ | $54_{\pm5}$ | $53_{\pm5}$ | $54_{\pm8}$ | $54_{\pm5}$ | $54_{\pm5}$ | $54_{\pm5}$ |
| D-VAT | $8_{\pm1}$ | $8_{\pm1}$ | $8_{\pm1}$ | $9_{\pm1}$ | $8_{\pm1}$ | $8_{\pm1}$ | $9_{\pm1}$ | $9_{\pm1}$ | $9_{\pm2}$ |
| **GC-VAT** | $\mathbf{209}_{\pm131}$ | $\mathbf{184}_{\pm136}$ | $\mathbf{202}_{\pm129}$ | $\mathbf{203}_{\pm119}$ | $\mathbf{189}_{\pm93}$ | $\mathbf{223}_{\pm114}$ | $\mathbf{167}_{\pm135}$ | $\mathbf{165}_{\pm126}$ | $\mathbf{178}_{\pm125}$ |
| Train: lake | citystreet | desert | village | downtown | lake | farmland | night | foggy | snow |
| AOT | $49_{\pm3}$ | $49_{\pm10}$ | $46_{\pm5}$ | $49_{\pm3}$ | $47_{\pm3}$ | $49_{\pm3}$ | $48_{\pm3}$ | $48_{\pm4}$ | $48_{\pm3}$ |
| D-VAT | $50_{\pm8}$ | $45_{\pm9}$ | $45_{\pm10}$ | $70_{\pm42}$ | $46_{\pm8}$ | $43_{\pm2}$ | $46_{\pm8}$ | $51_{\pm8}$ | $49_{\pm9}$ |
| **GC-VAT** | $\mathbf{112}_{\pm86}$ | $\mathbf{144}_{\pm110}$ | $\mathbf{203}_{\pm133}$ | $\mathbf{143}_{\pm134}$ | $\mathbf{181}_{\pm116}$ | $\mathbf{214}_{\pm111}$ | $\mathbf{190}_{\pm129}$ | $\mathbf{168}_{\pm110}$ | $\mathbf{99}_{\pm67}$ |
| Train: farmland | citystreet | desert | village | downtown | lake | farmland | night | foggy | snow |
| AOT | $51_{\pm7}$ | $50_{\pm9}$ | $46_{\pm5}$ | $49_{\pm3}$ | $51_{\pm9}$ | $60_{\pm25}$ | $48_{\pm4}$ | $56_{\pm24}$ | $56_{\pm24}$ |
| D-VAT | $13_{\pm2}$ | $13_{\pm1}$ | $13_{\pm1}$ | $15_{\pm1}$ | $14_{\pm1}$ | $13_{\pm1}$ | $14_{\pm1}$ | $13_{\pm1}$ | $14_{\pm1}$ |
| **GC-VAT** | $\mathbf{162}_{\pm89}$ | $\mathbf{170}_{\pm125}$ | $\mathbf{237}_{\pm128}$ | $\mathbf{81}_{\pm71}$ | $\mathbf{159}_{\pm119}$ | $\mathbf{243}_{\pm117}$ | $\mathbf{253}_{\pm109}$ | $\mathbf{245}_{\pm117}$ | $\mathbf{168}_{\pm105}$ |

Table 13: The detailed results of comparison experiments on $TSR$ metric.

| | Within / Cross Scene | | | | | | Cross Domain | | |
|---|---|---|---|---|---|---|---|---|---|
| Train: citystreet | citystreet | desert | village | downtown | lake | farmland | night | foggy | snow |
| AOT | $0.25_{\pm0.02}$ | $0.24_{\pm0.03}$ | $0.22_{\pm0.03}$ | $0.25_{\pm0.02}$ | $0.23_{\pm0.03}$ | $0.24_{\pm0.01}$ | $0.25_{\pm0.02}$ | $0.25_{\pm0.02}$ | $0.24_{\pm0.02}$ |
| D-VAT | $0.26_{\pm0.02}$ | $0.25_{\pm0.04}$ | $0.25_{\pm0.02}$ | $0.32_{\pm0.08}$ | $0.27_{\pm0.04}$ | $0.19_{\pm0.01}$ | $0.26_{\pm0.02}$ | $0.28_{\pm0.02}$ | $0.29_{\pm0.02}$ |
| **GC-VAT** | $\mathbf{0.80}_{\pm0.30}$ | $\mathbf{0.54}_{\pm0.32}$ | $\mathbf{0.50}_{\pm0.32}$ | $\mathbf{0.45}_{\pm0.30}$ | $\mathbf{0.44}_{\pm0.24}$ | $\mathbf{0.66}_{\pm0.27}$ | $\mathbf{0.72}_{\pm0.29}$ | $\mathbf{0.93}_{\pm0.14}$ | $\mathbf{0.79}_{\pm0.24}$ |
| Train: desert | citystreet | desert | village | downtown | lake | farmland | night | foggy | snow |
| AOT | $0.06_{\pm0.00}$ | $0.06_{\pm0.00}$ | $0.06_{\pm0.00}$ | $0.06_{\pm0.01}$ | $0.06_{\pm0.00}$ | $0.06_{\pm0.00}$ | $0.06_{\pm0.01}$ | $0.06_{\pm0.01}$ | $0.06_{\pm0.01}$ |
| D-VAT | $0.27_{\pm0.02}$ | $0.26_{\pm0.04}$ | $0.25_{\pm0.02}$ | $0.32_{\pm0.07}$ | $0.23_{\pm0.03}$ | $0.26_{\pm0.01}$ | $0.26_{\pm0.04}$ | $0.26_{\pm0.04}$ | $0.26_{\pm0.04}$ |
| **GC-VAT** | $\mathbf{0.73}_{\pm0.31}$ | $\mathbf{0.84}_{\pm0.29}$ | $\mathbf{0.87}_{\pm0.19}$ | $\mathbf{0.38}_{\pm0.32}$ | $\mathbf{0.56}_{\pm0.28}$ | $\mathbf{0.82}_{\pm0.25}$ | $\mathbf{0.57}_{\pm0.31}$ | $\mathbf{0.86}_{\pm0.28}$ | $\mathbf{0.86}_{\pm0.22}$ |
| Train: village | citystreet | desert | village | downtown | lake | farmland | night | foggy | snow |
| AOT | $0.25_{\pm0.03}$ | $0.25_{\pm0.04}$ | $0.23_{\pm0.03}$ | $0.24_{\pm0.02}$ | $0.25_{\pm0.02}$ | $0.26_{\pm0.06}$ | $0.23_{\pm0.03}$ | $0.23_{\pm0.03}$ | $0.23_{\pm0.03}$ |
| D-VAT | $0.23_{\pm0.04}$ | $0.23_{\pm0.04}$ | $0.22_{\pm0.05}$ | $0.31_{\pm0.14}$ | $0.24_{\pm0.06}$ | $0.22_{\pm0.01}$ | $0.22_{\pm0.04}$ | $0.22_{\pm0.05}$ | $0.23_{\pm0.05}$ |
| **GC-VAT** | $\mathbf{0.72}_{\pm0.28}$ | $\mathbf{0.51}_{\pm0.34}$ | $\mathbf{0.73}_{\pm0.32}$ | $\mathbf{0.46}_{\pm0.29}$ | $\mathbf{0.59}_{\pm0.33}$ | $\mathbf{0.48}_{\pm0.31}$ | $\mathbf{0.71}_{\pm0.32}$ | $\mathbf{0.71}_{\pm0.32}$ | $\mathbf{0.40}_{\pm0.29}$ |
| Train: downtown | citystreet | desert | village | downtown | lake | farmland | night | foggy | snow |
| AOT | $0.30_{\pm0.04}$ | $0.26_{\pm0.05}$ | $0.27_{\pm0.02}$ | $0.29_{\pm0.01}$ | $0.29_{\pm0.03}$ | $0.29_{\pm0.02}$ | $0.29_{\pm0.01}$ | $0.29_{\pm0.01}$ | $0.29_{\pm0.01}$ |
| D-VAT | $0.05_{\pm0.00}$ | $0.05_{\pm0.00}$ | $0.06_{\pm0.00}$ | $0.06_{\pm0.01}$ | $0.05_{\pm0.00}$ | $0.06_{\pm0.00}$ | $0.06_{\pm0.01}$ | $0.06_{\pm0.01}$ | $0.06_{\pm0.00}$ |
| **GC-VAT** | $\mathbf{0.77}_{\pm0.31}$ | $\mathbf{0.65}_{\pm0.30}$ | $\mathbf{0.67}_{\pm0.29}$ | $\mathbf{0.65}_{\pm0.30}$ | $\mathbf{0.49}_{\pm0.29}$ | $\mathbf{0.63}_{\pm0.33}$ | $\mathbf{0.58}_{\pm0.31}$ | $\mathbf{0.65}_{\pm0.29}$ | $\mathbf{0.64}_{\pm0.28}$ |
| Train: lake | citystreet | desert | village | downtown | lake | farmland | night | foggy | snow |
| AOT | $0.25_{\pm0.02}$ | $0.25_{\pm0.03}$ | $0.23_{\pm0.03}$ | $0.24_{\pm0.02}$ | $0.24_{\pm0.02}$ | $0.24_{\pm0.01}$ | $0.24_{\pm0.01}$ | $0.24_{\pm0.02}$ | $0.24_{\pm0.01}$ |
| D-VAT | $0.25_{\pm0.04}$ | $0.23_{\pm0.04}$ | $0.23_{\pm0.05}$ | $0.30_{\pm0.15}$ | $0.26_{\pm0.06}$ | $0.22_{\pm0.01}$ | $0.26_{\pm0.06}$ | $0.26_{\pm0.06}$ | $0.25_{\pm0.06}$ |
| **GC-VAT** | $\mathbf{0.43}_{\pm0.25}$ | $\mathbf{0.47}_{\pm0.30}$ | $\mathbf{0.64}_{\pm0.31}$ | $\mathbf{0.43}_{\pm0.28}$ | $\mathbf{0.61}_{\pm0.31}$ | $\mathbf{0.59}_{\pm0.30}$ | $\mathbf{0.59}_{\pm0.39}$ | $\mathbf{0.62}_{\pm0.32}$ | $\mathbf{0.41}_{\pm0.24}$ |
| Train: farmland | citystreet | desert | village | downtown | lake | farmland | night | foggy | snow |
| AOT | $0.24_{\pm0.02}$ | $0.24_{\pm0.04}$ | $0.22_{\pm0.03}$ | $0.25_{\pm0.02}$ | $0.24_{\pm0.02}$ | $0.23_{\pm0.01}$ | $0.23_{\pm0.01}$ | $0.23_{\pm0.01}$ | $0.23_{\pm0.01}$ |
| D-VAT | $0.07_{\pm0.01}$ | $0.07_{\pm0.01}$ | $0.07_{\pm0.00}$ | $0.08_{\pm0.01}$ | $0.07_{\pm0.00}$ | $0.07_{\pm0.00}$ | $0.08_{\pm0.00}$ | $0.07_{\pm0.00}$ | $0.08_{\pm0.00}$ |
| **GC-VAT** | $\mathbf{0.48}_{\pm0.24}$ | $\mathbf{0.59}_{\pm0.34}$ | $\mathbf{0.72}_{\pm0.26}$ | $\mathbf{0.33}_{\pm0.20}$ | $\mathbf{0.58}_{\pm0.28}$ | $\mathbf{0.68}_{\pm0.32}$ | $\mathbf{0.67}_{\pm0.32}$ | $\mathbf{0.78}_{\pm0.22}$ | $\mathbf{0.51}_{\pm0.28}$ |

**Sparse Reward.** In addition to the dense reward function described in the main text, we also provide a sparse reward function design. The sparse reward only provides a fixed reward when the target is within the image and no reward when it is outside. The definition of $r_d$ is as follows.

$$r_d = \begin{cases} 1, & t \in \mathcal{I} \\ 0, & \text{otherwise} \end{cases}, \tag{14}$$

where $\mathcal{I}$ represents the image range. This reward can be used to construct the metric, Tracking Success Rate (TSR).

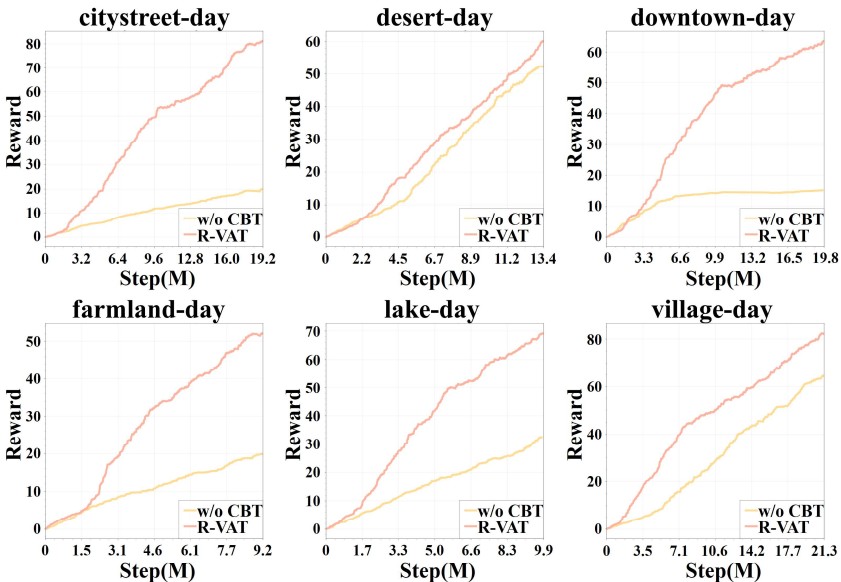

Figure 13: Schematic diagram of reward curves on DAT scenes.

**Training algorithm.** For the training method of GC-VAT, we choose to use PPO algorithm. PPO algorithm regulates the speed of gradient updates by constraining the magnitude of policy changes $r_t$, expressed as follows:

$$r_t(\theta) = \frac{\pi_\theta(a_t|s_t)}{\pi_{\theta_{old}}(a_t|s_t)}, \tag{15}$$

where $\pi_\theta$ and $\pi_{\theta_{old}}$ are the new and old policies. Additionally, to enhance the agent's exploration, we introduce an entropy loss term $\mathcal{H}$, formulated as:

$$\mathcal{H}(\pi_\theta(s)) = -\sum_a \pi_\theta(a|s)\log \pi_\theta(a|s). \tag{16}$$

The optimization objective for the actor is as follows:

$$\mathcal{L}_A = \hat{\mathbb{E}}[\min\left(r_t\hat{A}_t, \text{clip}(r_t, 1-\epsilon, 1+\epsilon)\hat{A}_t\right)+\beta\mathcal{H}], \tag{17}$$

where $\hat{A}_t$ is the advantage function, $\epsilon$ is the clip parameter, and $\beta$ is the entropy coefficient. The expression of $\hat{A}_t$ is:

$$\hat{A}_t = \sum_{l=0}^{E_l-t} (\gamma\lambda)^l\delta_{t+l}, \tag{18}$$

where $T, \lambda, \delta_{t+l}$ are the data collection step, generalized advantage estimator (GAE) [56] discount factor and temporal difference error respectively. The optimization objective expression of the critic network $V$ is defined as:

$$\mathcal{L}_C = \hat{\mathbb{E}}_t[(r_t + \gamma V(s_{t+1}) - V(s_t))^2]. \tag{19}$$

The hyperparameters of the PPO algorithm used in this article are set as follows: discount factor $\gamma = 0.9$, GAE discount factor $\lambda = 0.95$, entropy coefficient $\beta = 0.01$, PPO clipping parameter $\epsilon = 0.2$.

**Curriculum Learning for Agent Training.** We introduce a Curriculum-Based Training (CBT) strategy designed to progressively enhance the performance of the tracker. In the first-stage curriculum, the agent is trained to track vehicles moving along straight trajectories without occlusions or extra interference. In the second-stage curriculum, the agent is exposed to visually complex environments and tasked with tracking targets exhibiting diverse and dynamic behaviors. The scenario of each stage is shown in Fig. 12, where the upper row is the first-stage environment, and the lower row corresponds to the second-stage environment.

Table 14: Effectiveness of CBT strategy on the DAT benchmark, results from $CR$ metric.

| | Within / Cross Scene | | | | | | Cross Domain | | |
|---|---|---|---|---|---|---|---|---|---|
| Train: citystreet | citystreet | desert | village | downtown | lake | farmland | night | foggy | snow |
| w/o CBT | $54_{\pm7}$ | $37_{\pm21}$ | $30_{\pm6}$ | $30_{\pm14}$ | $48_{\pm13}$ | $48_{\pm4}$ | $54_{\pm9}$ | $54_{\pm9}$ | $54_{\pm9}$ |
| GC-VAT | $\mathbf{279}_{\pm110}$ | $\mathbf{129}_{\pm112}$ | $\mathbf{153}_{\pm119}$ | $\mathbf{135}_{\pm109}$ | $\mathbf{112}_{\pm92}$ | $\mathbf{191}_{\pm122}$ | $\mathbf{257}_{\pm126}$ | $\mathbf{316}_{\pm84}$ | $\mathbf{202}_{\pm119}$ |
| Train: desert | citystreet | desert | village | downtown | lake | farmland | night | foggy | snow |
| w/o CBT | $253_{\pm132}$ | $302_{\pm99}$ | $284_{\pm92}$ | $175_{\pm102}$ | $236_{\pm123}$ | $266_{\pm110}$ | $241_{\pm127}$ | $279_{\pm120}$ | $306_{\pm95}$ |
| GC-VAT | $\mathbf{278}_{\pm111}$ | $\mathbf{307}_{\pm124}$ | $\mathbf{305}_{\pm94}$ | $119_{\pm110}$ | $170_{\pm139}$ | $\mathbf{275}_{\pm121}$ | $182_{\pm131}$ | $\mathbf{307}_{\pm124}$ | $\mathbf{307}_{\pm97}$ |
| Train: village | citystreet | desert | village | downtown | lake | farmland | night | foggy | snow |
| w/o CBT | $230_{\pm120}$ | $\mathbf{197}_{\pm124}$ | $\mathbf{255}_{\pm118}$ | $59_{\pm69}$ | $126_{\pm105}$ | $\mathbf{182}_{\pm120}$ | $\mathbf{267}_{\pm93}$ | $208_{\pm141}$ | $73_{\pm68}$ |
| GC-VAT | $\mathbf{234}_{\pm122}$ | $160_{\pm139}$ | $239_{\pm134}$ | $\mathbf{93}_{\pm102}$ | $\mathbf{153}_{\pm115}$ | $140_{\pm118}$ | $257_{\pm122}$ | $\mathbf{257}_{\pm120}$ | $\mathbf{114}_{\pm115}$ |
| Train: downtown | citystreet | desert | village | downtown | lake | farmland | night | foggy | snow |
| w/o CBT | $54_{\pm9}$ | $49_{\pm13}$ | $47_{\pm8}$ | $57_{\pm15}$ | $51_{\pm9}$ | $48_{\pm4}$ | $29_{\pm3}$ | $57_{\pm15}$ | $58_{\pm15}$ |
| GC-VAT | $\mathbf{209}_{\pm131}$ | $\mathbf{184}_{\pm136}$ | $\mathbf{202}_{\pm129}$ | $\mathbf{203}_{\pm119}$ | $\mathbf{189}_{\pm93}$ | $\mathbf{223}_{\pm114}$ | $\mathbf{167}_{\pm135}$ | $\mathbf{165}_{\pm126}$ | $\mathbf{178}_{\pm125}$ |
| Train: lake | citystreet | desert | village | downtown | lake | farmland | night | foggy | snow |
| w/o CBT | $\mathbf{124}_{\pm90}$ | $88_{\pm52}$ | $191_{\pm108}$ | $93_{\pm75}$ | $\mathbf{187}_{\pm123}$ | $198_{\pm117}$ | $183_{\pm110}$ | $\mathbf{185}_{\pm102}$ | $\mathbf{102}_{\pm57}$ |
| GC-VAT | $112_{\pm86}$ | $\mathbf{144}_{\pm110}$ | $\mathbf{203}_{\pm133}$ | $\mathbf{143}_{\pm134}$ | $181_{\pm116}$ | $\mathbf{214}_{\pm111}$ | $\mathbf{190}_{\pm129}$ | $168_{\pm110}$ | $99_{\pm67}$ |
| Train: farmland | citystreet | desert | village | downtown | lake | farmland | night | foggy | snow |
| w/o CBT | $52_{\pm9}$ | $47_{\pm9}$ | $45_{\pm9}$ | $69_{\pm42}$ | $50_{\pm9}$ | $46_{\pm2}$ | $46_{\pm2}$ | $46_{\pm3}$ | $46_{\pm2}$ |
| GC-VAT | $\mathbf{162}_{\pm89}$ | $\mathbf{170}_{\pm125}$ | $\mathbf{237}_{\pm128}$ | $\mathbf{81}_{\pm71}$ | $\mathbf{159}_{\pm119}$ | $\mathbf{243}_{\pm117}$ | $\mathbf{253}_{\pm109}$ | $\mathbf{245}_{\pm117}$ | $\mathbf{168}_{\pm105}$ |

Table 15: Effectiveness of CBT strategy on the DAT benchmark, results from $TSR$ metric.

| | Within / Cross Scene | | | | | | Cross Domain | | |
|---|---|---|---|---|---|---|---|---|---|
| Train: citystreet | citystreet | desert | village | downtown | lake | farmland | night | foggy | snow |
| w/o CBT | $0.30_{\pm0.05}$ | $0.14_{\pm0.10}$ | $0.20_{\pm0.10}$ | $0.31_{\pm0.15}$ | $0.28_{\pm0.06}$ | $0.21_{\pm0.01}$ | $0.30_{\pm0.05}$ | $0.30_{\pm0.05}$ | $0.30_{\pm0.05}$ |
| GC-VAT | $\mathbf{0.80}_{\pm0.30}$ | $\mathbf{0.54}_{\pm0.32}$ | $\mathbf{0.50}_{\pm0.32}$ | $\mathbf{0.45}_{\pm0.30}$ | $\mathbf{0.44}_{\pm0.24}$ | $\mathbf{0.66}_{\pm0.27}$ | $\mathbf{0.72}_{\pm0.29}$ | $\mathbf{0.93}_{\pm0.14}$ | $\mathbf{0.79}_{\pm0.24}$ |
| Train: desert | citystreet | desert | village | downtown | lake | farmland | night | foggy | snow |
| w/o CBT | $\mathbf{0.83}_{\pm0.28}$ | $0.75_{\pm0.32}$ | $0.66_{\pm0.34}$ | $\mathbf{0.52}_{\pm0.28}$ | $\mathbf{0.69}_{\pm0.24}$ | $0.74_{\pm0.26}$ | $\mathbf{0.59}_{\pm0.36}$ | $0.74_{\pm0.34}$ | $0.75_{\pm0.34}$ |
| GC-VAT | $0.73_{\pm0.31}$ | $\mathbf{0.84}_{\pm0.29}$ | $\mathbf{0.87}_{\pm0.19}$ | $0.38_{\pm0.32}$ | $0.56_{\pm0.28}$ | $\mathbf{0.82}_{\pm0.25}$ | $0.57_{\pm0.31}$ | $\mathbf{0.86}_{\pm0.28}$ | $\mathbf{0.86}_{\pm0.22}$ |
| Train: village | citystreet | desert | village | downtown | lake | farmland | night | foggy | snow |
| w/o CBT | $\mathbf{0.73}_{\pm0.28}$ | $\mathbf{0.62}_{\pm0.28}$ | $\mathbf{0.82}_{\pm0.16}$ | $0.23_{\pm0.17}$ | $0.46_{\pm0.25}$ | $\mathbf{0.58}_{\pm0.33}$ | $\mathbf{0.71}_{\pm0.28}$ | $0.69_{\pm0.33}$ | $\mathbf{0.40}_{\pm0.24}$ |
| GC-VAT | $0.72_{\pm0.28}$ | $0.51_{\pm0.34}$ | $0.73_{\pm0.32}$ | $\mathbf{0.46}_{\pm0.29}$ | $\mathbf{0.59}_{\pm0.33}$ | $0.48_{\pm0.31}$ | $0.71_{\pm0.32}$ | $\mathbf{0.71}_{\pm0.32}$ | $0.40_{\pm0.29}$ |
| Train: downtown | citystreet | desert | village | downtown | lake | farmland | night | foggy | snow |
| w/o CBT | $0.29_{\pm0.04}$ | $0.27_{\pm0.03}$ | $0.27_{\pm0.03}$ | $0.33_{\pm0.06}$ | $0.28_{\pm0.03}$ | $0.27_{\pm0.01}$ | $0.33_{\pm0.06}$ | $0.33_{\pm0.06}$ | $0.33_{\pm0.06}$ |
| GC-VAT | $\mathbf{0.77}_{\pm0.31}$ | $\mathbf{0.65}_{\pm0.30}$ | $\mathbf{0.67}_{\pm0.29}$ | $\mathbf{0.65}_{\pm0.30}$ | $\mathbf{0.49}_{\pm0.29}$ | $\mathbf{0.63}_{\pm0.33}$ | $\mathbf{0.58}_{\pm0.31}$ | $\mathbf{0.65}_{\pm0.29}$ | $\mathbf{0.64}_{\pm0.28}$ |
| Train: lake | citystreet | desert | village | downtown | lake | farmland | night | foggy | snow |
| w/o CBT | $\mathbf{0.51}_{\pm0.30}$ | $\mathbf{0.47}_{\pm0.29}$ | $0.45_{\pm0.22}$ | $\mathbf{0.44}_{\pm0.23}$ | $0.57_{\pm0.28}$ | $\mathbf{0.59}_{\pm0.26}$ | $\mathbf{0.78}_{\pm0.22}$ | $\mathbf{0.62}_{\pm0.24}$ | $0.33_{\pm0.15}$ |
| GC-VAT | $0.43_{\pm0.25}$ | $0.47_{\pm0.30}$ | $\mathbf{0.64}_{\pm0.31}$ | $0.43_{\pm0.28}$ | $\mathbf{0.61}_{\pm0.31}$ | $\mathbf{0.59}_{\pm0.30}$ | $0.59_{\pm0.39}$ | $0.62_{\pm0.32}$ | $\mathbf{0.41}_{\pm0.24}$ |
| Train: farmland | citystreet | desert | village | downtown | lake | farmland | night | foggy | snow |
| w/o CBT | $0.26_{\pm0.04}$ | $0.24_{\pm0.04}$ | $0.23_{\pm0.05}$ | $0.31_{\pm0.14}$ | $0.26_{\pm0.06}$ | $0.23_{\pm0.01}$ | $0.23_{\pm0.01}$ | $0.23_{\pm0.01}$ | $0.23_{\pm0.01}$ |
| GC-VAT | $\mathbf{0.48}_{\pm0.24}$ | $\mathbf{0.59}_{\pm0.34}$ | $\mathbf{0.72}_{\pm0.26}$ | $\mathbf{0.33}_{\pm0.20}$ | $\mathbf{0.58}_{\pm0.28}$ | $\mathbf{0.68}_{\pm0.32}$ | $\mathbf{0.67}_{\pm0.32}$ | $\mathbf{0.78}_{\pm0.22}$ | $\mathbf{0.51}_{\pm0.28}$ |

## D  Baselines

**Active Object Tracking (AOT)** [39]. In this paper, the agent learns to follow a fixed target-tracking trajectory using A3C. In addition, the agent uses the following reward:

$$r = A - (\frac{\sqrt{x^2 + (y - d)^2}}{c} + \lambda \mid \omega \mid), \tag{20}$$

where $d$ represents the optimal distance between the tracker and the target, $c$ is the maximum allowable distance, and $A$ denotes the maximum reward. In the original paper, $c = 200$ and $A = 1.0$. During our replication, we set $A = 1.0$, but due to the drone's camera being tilted downward, a value of $c = 200$ would far exceed the camera's field of view, which is unrealistic. Therefore, we modify the parameter $c$ to be the maximum offset distance that keeps the target within the image, i.e., $c = 9$.

**D-VAT**[19]. In this approach, the agent uses an asymmetric Actor-Critic network structure and the soft actor-critic learning method [25] to accomplish the task of drone tracking another drone. In the

Table 16: Effectiveness of reward design on the DAT benchmark, results from $CR$ metric.

| | Within / Cross Scene | | | | | | Cross Domain | | |
|---|---|---|---|---|---|---|---|---|---|
| Train: citystreet | citystreet | desert | village | downtown | lake | farmland | night | foggy | snow |
| $R_{\text{D-VAT}}$ | $9_{\pm 1}$ | $8_{\pm 1}$ | $8_{\pm 0}$ | $8_{\pm 1}$ | $9_{\pm 0}$ | $9_{\pm 0}$ | $9_{\pm 1}$ | $9_{\pm 1}$ | $9_{\pm 1}$ |
| **GC-VAT** | $\mathbf{279}_{\pm 110}$ | $\mathbf{129}_{\pm 112}$ | $\mathbf{153}_{\pm 119}$ | $\mathbf{135}_{\pm 109}$ | $\mathbf{112}_{\pm 92}$ | $\mathbf{191}_{\pm 122}$ | $\mathbf{257}_{\pm 126}$ | $\mathbf{316}_{\pm 84}$ | $\mathbf{202}_{\pm 119}$ |
| Train: desert | citystreet | desert | village | downtown | lake | farmland | night | foggy | snow |
| $R_{\text{D-VAT}}$ | $9_{\pm 1}$ | $9_{\pm 0}$ | $8_{\pm 1}$ | $9_{\pm 0}$ | $8_{\pm 0}$ | $10_{\pm 0}$ | $8_{\pm 1}$ | $10_{\pm 1}$ | $8_{\pm 0}$ |
| **GC-VAT** | $\mathbf{278}_{\pm 111}$ | $\mathbf{307}_{\pm 124}$ | $\mathbf{305}_{\pm 94}$ | $\mathbf{119}_{\pm 110}$ | $\mathbf{170}_{\pm 139}$ | $\mathbf{275}_{\pm 121}$ | $\mathbf{182}_{\pm 131}$ | $\mathbf{307}_{\pm 124}$ | $\mathbf{307}_{\pm 97}$ |
| Train: village | citystreet | desert | village | downtown | lake | farmland | night | foggy | snow |
| $R_{\text{D-VAT}}$ | $9_{\pm 1}$ | $8_{\pm 1}$ | $9_{\pm 1}$ | $9_{\pm 1}$ | $8_{\pm 1}$ | $9_{\pm 0}$ | $8_{\pm 1}$ | $8_{\pm 1}$ | $8_{\pm 1}$ |
| **GC-VAT** | $\mathbf{234}_{\pm 122}$ | $\mathbf{160}_{\pm 139}$ | $\mathbf{239}_{\pm 134}$ | $\mathbf{93}_{\pm 102}$ | $\mathbf{153}_{\pm 115}$ | $\mathbf{140}_{\pm 118}$ | $\mathbf{257}_{\pm 122}$ | $\mathbf{257}_{\pm 120}$ | $\mathbf{114}_{\pm 115}$ |
| Train: downtown | citystreet | desert | village | downtown | lake | farmland | night | foggy | snow |
| $R_{\text{D-VAT}}$ | $8_{\pm 1}$ | $8_{\pm 0}$ | $8_{\pm 1}$ | $9_{\pm 1}$ | $8_{\pm 1}$ | $8_{\pm 1}$ | $9_{\pm 1}$ | $9_{\pm 1}$ | $9_{\pm 0}$ |
| **GC-VAT** | $\mathbf{209}_{\pm 131}$ | $\mathbf{184}_{\pm 136}$ | $\mathbf{202}_{\pm 129}$ | $\mathbf{203}_{\pm 119}$ | $\mathbf{189}_{\pm 93}$ | $\mathbf{223}_{\pm 114}$ | $\mathbf{167}_{\pm 135}$ | $\mathbf{165}_{\pm 126}$ | $\mathbf{178}_{\pm 125}$ |
| Train: lake | citystreet | desert | village | downtown | lake | farmland | night | foggy | snow |
| $R_{\text{D-VAT}}$ | $11_{\pm 3}$ | $11_{\pm 1}$ | $9_{\pm 1}$ | $9_{\pm 2}$ | $9_{\pm 0}$ | $8_{\pm 0}$ | $9_{\pm 0}$ | $10_{\pm 1}$ | $8_{\pm 1}$ |
| **GC-VAT** | $\mathbf{112}_{\pm 86}$ | $\mathbf{144}_{\pm 110}$ | $\mathbf{203}_{\pm 133}$ | $\mathbf{143}_{\pm 134}$ | $\mathbf{181}_{\pm 116}$ | $\mathbf{214}_{\pm 111}$ | $\mathbf{190}_{\pm 129}$ | $\mathbf{168}_{\pm 110}$ | $\mathbf{99}_{\pm 67}$ |
| Train: farmland | citystreet | desert | village | downtown | lake | farmland | night | foggy | snow |
| $R_{\text{D-VAT}}$ | $9_{\pm 1}$ | $8_{\pm 1}$ | $8_{\pm 1}$ | $9_{\pm 1}$ | $8_{\pm 1}$ | $9_{\pm 1}$ | $9_{\pm 0}$ | $9_{\pm 0}$ | $9_{\pm 0}$ |
| **GC-VAT** | $\mathbf{162}_{\pm 89}$ | $\mathbf{170}_{\pm 125}$ | $\mathbf{237}_{\pm 128}$ | $\mathbf{81}_{\pm 71}$ | $\mathbf{159}_{\pm 119}$ | $\mathbf{243}_{\pm 117}$ | $\mathbf{253}_{\pm 109}$ | $\mathbf{245}_{\pm 117}$ | $\mathbf{168}_{\pm 105}$ |

Table 17: Effectiveness of reward design on the DAT benchmark, results from $TSR$ metric.

| | Within / Cross Scene | | | | | | Cross Domain | | |
|---|---|---|---|---|---|---|---|---|---|
| Train: citystreet | citystreet | desert | village | downtown | lake | farmland | night | foggy | snow |
| $R_{\text{D-VAT}}$ | $0.06_{\pm 0.00}$ | $0.05_{\pm 0.00}$ | $0.06_{\pm 0.00}$ | $0.06_{\pm 0.00}$ | $0.06_{\pm 0.00}$ | $0.06_{\pm 0.00}$ | $0.06_{\pm 0.00}$ | $0.06_{\pm 0.01}$ | $0.06_{\pm 0.00}$ |
| **GC-VAT** | $\mathbf{0.80}_{\pm 0.30}$ | $\mathbf{0.54}_{\pm 0.32}$ | $\mathbf{0.50}_{\pm 0.32}$ | $\mathbf{0.45}_{\pm 0.30}$ | $\mathbf{0.44}_{\pm 0.24}$ | $\mathbf{0.66}_{\pm 0.27}$ | $\mathbf{0.72}_{\pm 0.29}$ | $\mathbf{0.93}_{\pm 0.14}$ | $\mathbf{0.79}_{\pm 0.24}$ |
| Train: desert | citystreet | desert | village | downtown | lake | farmland | night | foggy | snow |
| $R_{\text{D-VAT}}$ | $0.06_{\pm 0.00}$ | $0.06_{\pm 0.00}$ | $0.06_{\pm 0.00}$ | $0.06_{\pm 0.01}$ | $0.06_{\pm 0.00}$ | $0.10_{\pm 0.00}$ | $0.06_{\pm 0.00}$ | $0.09_{\pm 0.01}$ | $0.06_{\pm 0.00}$ |
| **GC-VAT** | $\mathbf{0.73}_{\pm 0.31}$ | $\mathbf{0.84}_{\pm 0.29}$ | $\mathbf{0.87}_{\pm 0.19}$ | $\mathbf{0.38}_{\pm 0.32}$ | $0.56_{\pm 0.28}$ | $\mathbf{0.82}_{\pm 0.25}$ | $\mathbf{0.57}_{\pm 0.31}$ | $\mathbf{0.86}_{\pm 0.28}$ | $\mathbf{0.86}_{\pm 0.22}$ |
| Train: village | citystreet | desert | village | downtown | lake | farmland | night | foggy | snow |
| $R_{\text{D-VAT}}$ | $0.06_{\pm 0.01}$ | $0.06_{\pm 0.00}$ | $0.06_{\pm 0.00}$ | $0.06_{\pm 0.01}$ | $0.05_{\pm 0.00}$ | $0.06_{\pm 0.00}$ | $0.05_{\pm 0.00}$ | $0.06_{\pm 0.00}$ | $0.06_{\pm 0.00}$ |
| **GC-VAT** | $\mathbf{0.72}_{\pm 0.28}$ | $\mathbf{0.51}_{\pm 0.34}$ | $\mathbf{0.73}_{\pm 0.32}$ | $\mathbf{0.46}_{\pm 0.29}$ | $\mathbf{0.59}_{\pm 0.33}$ | $\mathbf{0.48}_{\pm 0.31}$ | $\mathbf{0.71}_{\pm 0.32}$ | $\mathbf{0.71}_{\pm 0.32}$ | $\mathbf{0.40}_{\pm 0.29}$ |
| Train: downtown | citystreet | desert | village | downtown | lake | farmland | night | foggy | snow |
| $R_{\text{D-VAT}}$ | $0.06_{\pm 0.01}$ | $0.06_{\pm 0.00}$ | $0.06_{\pm 0.00}$ | $0.06_{\pm 0.00}$ | $0.05_{\pm 0.00}$ | $0.06_{\pm 0.00}$ | $0.06_{\pm 0.00}$ | $0.06_{\pm 0.00}$ | $0.06_{\pm 0.00}$ |
| **GC-VAT** | $\mathbf{0.77}_{\pm 0.31}$ | $\mathbf{0.65}_{\pm 0.30}$ | $\mathbf{0.67}_{\pm 0.29}$ | $\mathbf{0.65}_{\pm 0.30}$ | $\mathbf{0.49}_{\pm 0.29}$ | $\mathbf{0.63}_{\pm 0.33}$ | $\mathbf{0.58}_{\pm 0.31}$ | $\mathbf{0.65}_{\pm 0.29}$ | $\mathbf{0.64}_{\pm 0.28}$ |
| Train: lake | citystreet | desert | village | downtown | lake | farmland | night | foggy | snow |
| $R_{\text{D-VAT}}$ | $0.10_{\pm 0.01}$ | $0.09_{\pm 0.01}$ | $0.07_{\pm 0.00}$ | $0.06_{\pm 0.01}$ | $0.06_{\pm 0.00}$ | $0.06_{\pm 0.00}$ | $0.06_{\pm 0.00}$ | $0.08_{\pm 0.00}$ | $0.06_{\pm 0.00}$ |
| **GC-VAT** | $\mathbf{0.43}_{\pm 0.25}$ | $\mathbf{0.47}_{\pm 0.30}$ | $\mathbf{0.64}_{\pm 0.31}$ | $\mathbf{0.43}_{\pm 0.28}$ | $\mathbf{0.61}_{\pm 0.31}$ | $\mathbf{0.59}_{\pm 0.30}$ | $\mathbf{0.59}_{\pm 0.39}$ | $\mathbf{0.62}_{\pm 0.32}$ | $\mathbf{0.41}_{\pm 0.24}$ |
| Train: farmland | citystreet | desert | village | downtown | lake | farmland | night | foggy | snow |
| $R_{\text{D-VAT}}$ | $0.06_{\pm 0.00}$ | $0.05_{\pm 0.00}$ | $0.05_{\pm 0.00}$ | $0.06_{\pm 0.01}$ | $0.05_{\pm 0.00}$ | $0.06_{\pm 0.00}$ | $0.06_{\pm 0.00}$ | $0.06_{\pm 0.00}$ | $0.06_{\pm 0.00}$ |
| **GC-VAT** | $\mathbf{0.48}_{\pm 0.24}$ | $\mathbf{0.59}_{\pm 0.34}$ | $\mathbf{0.72}_{\pm 0.26}$ | $\mathbf{0.33}_{\pm 0.20}$ | $\mathbf{0.58}_{\pm 0.28}$ | $\mathbf{0.68}_{\pm 0.32}$ | $\mathbf{0.67}_{\pm 0.32}$ | $\mathbf{0.78}_{\pm 0.22}$ | $\mathbf{0.51}_{\pm 0.28}$ |

actual comparative experiments, we convert it from a continuous action space to a discrete action space, referring to [10]. Additionally, the method uses the following reward function.

$$r(k) = \begin{cases} r_e(k) - k_v r_v(k) - k_u r_u(k) & \|y(k)\| > d_m \\ -k_c & \text{otherwise,} \end{cases} \qquad (21)$$

In the above equation Eq. 21, $r_v(k)$ and $r_u(k)$ are regularization terms for the drone's speed and output control, as shown in Eq. 22. For the discrete action space, the regularization term has a fixed value for a given action. This term only regularizes the linear velocity of the drone, which causes the drone to tend to perform rotational movements. Therefore, in the reproduction process, we set $k_v = 0$ and $k_u = 0$. Additionally, due to the unexpectedly large acceleration values obtained for the target relative to the tracker under the discrete action setting, we set the input acceleration of the critic

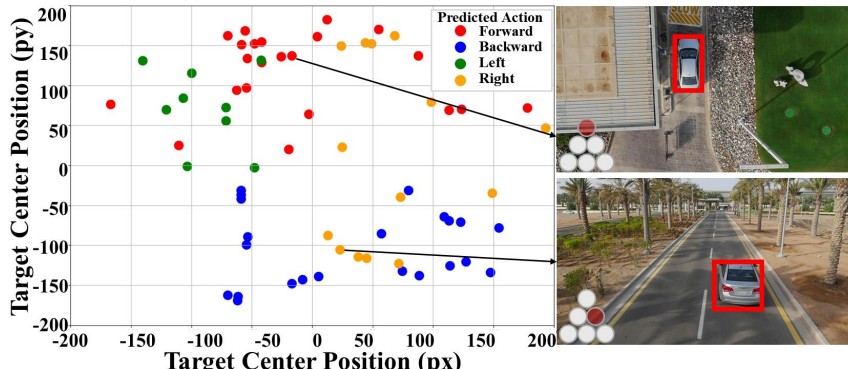

Figure 14: Qualitative results on images from the *car6* video sequence. Arrows link data points to the visualization of associated scenarios.

network to $a(k) = 0$.

$$r_v(k) = \frac{\|v(k)\|}{1 + \|v(k)\|}, \quad r_u(k) = \frac{\|u(k)\|}{1 + \|u(k)\|}. \tag{22}$$

It is important to note that in the AOT and D-VAT experiments, the target is initially positioned at the center of the tracker's image, and the initial forward directions of both the tracker and the target are aligned. Additionally, since the success criterion of DAT requires the agent to keep the target at the center of its view, the optimal distance between the tracker and the target is defined as the distance in the forward direction when the target is at the center of the camera's field of view. The tracker's flight altitude is set to 22 meters, and the gimbal pitch angle is 1.37 radians, which remains consistent with the parameters used during testing.

# E    More Experiments

## E.1    Experiment Settings

**More Implementation Details.** The training involves a range of 9.2M to 21.3M steps across 35 parallel environments. The webots runs at 500Hz, with the algorithm updating every four steps (125Hz). Episodes last up to 1500 steps and were terminated early if the drone lost the target for over 100 consecutive steps, collided, or crashed. The drone translation speed is set to 40m/s, and rotational speed to 2rad/s. The map features 40 vehicles, each with a maximum speed of 20m/s and acceleration of $\pm 25\text{m/s}^2$. During testing, the altitude is set to 22m, the pitch angle to 1.37rad, and the target initializes at the camera's center.

The drone's translation speed is set to a higher value to prevent it from becoming too similar to the target's speed (with a maximum of 20 m/s). This prevents simple forward movement from yielding excessively high reward evaluations. If the drone's speed is set lower (e.g., 20 m/s), it may adopt a suboptimal strategy, relying solely on one action.

Due to the varying challenges posed by different scene maps, the convergence speed of the agent differs across experiments. The training steps are shown in Table 10.

**Ablation Experiment Settings.** In this section, we introduce the training conditions of the single-stage RL and GC-VAT, as well as the criteria for stage transitions. In single-stage RL, the agent is placed in one of six scenarios (*citystreet*, *desert*, *village*, *downtown*, *lake*, and *farmland*) for training. For GC-VAT, the agent is first trained in an environment where a randomly colored target moves straight along a line without obstacles. After convergence, the model is then trained in the corresponding complex scenarios. The transition steps **T** for GC-VAT are in Table 11.

## E.2    Comparison Experiments

We provide a comprehensive analysis of the comparative experimental results.

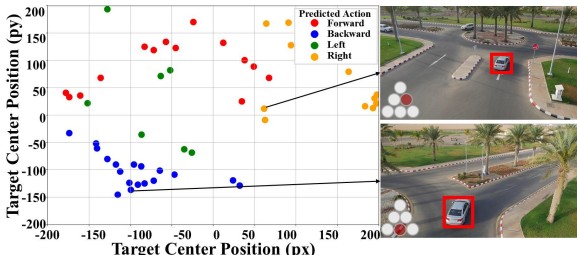

Figure 15: Qualitative results on the *car8* video.

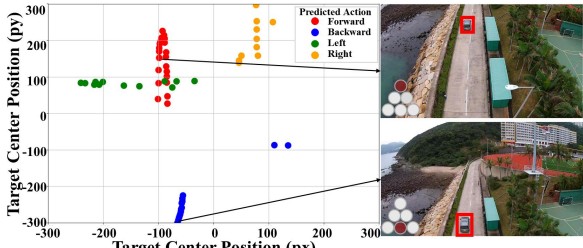

Figure 16: Qualitative results on the *Car4* video.

Table 18: Results (metric is Correct Action Rate) of 8 videos in VOT benchmark.

| Video | car1 | car3 | car6 | car8 |
|---|---|---|---|---|
| Random | 0.418 | 0.434 | 0.418 | 0.430 |
| **Ours** | **0.696** | **0.845** | **0.754** | **0.833** |
| Video | carchase | car16 | following | car9 |
| Random | 0.429 | 0.421 | 0.314 | 0.439 |
| **Ours** | **0.870** | **0.834** | **0.773** | **0.756** |

Table 19: Results of 8 videos in DTB70 benchmark.

| Video | Car2 | Car4 | Car5 | RcCar4 |
|---|---|---|---|---|
| Random | 0.419 | 0.421 | 0.429 | 0.436 |
| **Ours** | **0.757** | **0.894** | **0.893** | **0.876** |
| Video | Car8 | RaceCar | RaceCar1 | RcCar3 |
| Random | 0.411 | 0.462 | 0.430 | 0.400 |
| **Ours** | **0.803** | **0.713** | **0.880** | **0.851** |

Specifically, we provide detailed evaluations for within-scene (same scenes, same weather), cross-scene and cross-domain testing. Table 12 reports the $CR$ metric of three models under cross-scene and cross-domain conditions, while Table 13 presents the $TSR$ metric.

As shown in Table 12 and Table 13, the proposed GC-VAT significantly outperforms SOTA methods. Due to the reward design based on physical distance, both the AOT [39] and D-VAT [19] fail to accurately reflect the agent's tracking performance from a top-down perspective (see Appendix C.1 for theoretical proof), leading to misleading training signals for the tracker. Consequently, neither AOT nor D-VAT can effectively learn meaningful features, resulting in irregular performance distributions. In contrast, the proposed GC-VAT achieves superior convergence across all scenes. Specifically, in cross-scene experiments, the testing performance of the agent on the *downtown* map is relatively low, indicating that dense buildings and complex road elements pose significant challenges to the agent. Conversely, the testing performance on the *village* map is comparatively high, suggesting that the uniform color and simpler road conditions in the village map present fewer challenges.

For cross-domain testing experiments, the agent performs well under *night* and *foggy* conditions but struggles under *snow* conditions. This indicates that the proposed GC-VAT exhibits strong robustness to changes in lighting and visibility but is less adaptive to variations in scene tone.

### E.3 Ablation Experiments

We present a comprehensive analysis of the ablation studies. First, we provide the reward curves for the *citystreet*, *desert*, *village*, *downtown*, *lake*, and *farmland* maps (see Fig. 13). Next, we provide detailed experimental results on the effectiveness of the Curriculum-Based Training strategy, as shown in Table 14 and Table 15.

Finally, the effectiveness of the reward in the GC-VAT can be found in Table 16 and Table 17.

**Effectiveness of reward design.** To experimentally validate the effectiveness of the reward design proposed in this paper and to corroborate the theoretical proof in Appendix C.1, we conduct ablation experiments on the reward function. The comparative method utilizes the reward function from [19]. The detailed experimental results for the $CR$ and $TSR$ metrics are provided in Table 16 and Table 17. For within-scene testing, the GC-VAT achieves an average improvement of $1100\%(0.06 \rightarrow 0.72)$ in the $TSR$ metric compared to the reward design in [19]. In cross-scene and cross-domain testing, the GC-VAT achieves average enhancements of $850\%(0.06 \rightarrow 0.57)$ and $1017\%(0.06 \rightarrow 0.67)$ in the $TSR$ metric, respectively. These results demonstrate the high effectiveness of the proposed reward.

**Effectiveness of Curriculum-Based Training strategy.** To validate the effectiveness of the proposed Curriculum-Based Training (CBT) strategy, we conduct ablation experiments by removing the CBT module. The results for the $CR$ and $TSR$ metrics are presented in Table 14 and Table 15, respectively.

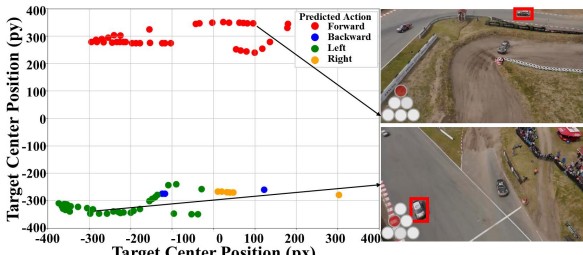

Figure 17: Qualitative results on the *RaceCar1* video.

Table 20: Results of 8 videos in UAVDT [20] benchmark.

| Video | S1603 | S0201 | S0101 | S0306 |
|---|---|---|---|---|
| Random | 0.435 | 0.438 | 0.422 | 0.437 |
| **Ours** | **0.896** | **0.806** | **0.865** | **0.773** |
| **Video** | *S1201* | *S0303* | *S1301* | *S1701* |
| Random | 0.445 | 0.385 | 0.397 | 0.407 |
| **Ours** | **0.867** | **0.735** | **0.760** | **0.713** |

The experimental results demonstrate that single-stage reinforcement learning methods without the CBT strategy successfully learn task objectives and achieve convergence on the *desert*, *village*, and *lake* maps. These three maps exhibit similar environmental characteristics: the *desert* and *village* maps feature uniform background colors and relatively simple road elements. Although the *desert* map has road segments partially covered by sand, these challenges are easy for the agent to overcome. Similarly, while the *village* map includes tunnels that may block vision, the proportion of tunnels is low. Additionally, although the *lake* map exhibits diverse background colors, the diversity primarily arises from vegetation-covered areas, which occupy a small proportion of the map, resulting in low challenges for the agent. In contrast, single-stage reinforcement learning methods without the CBT strategy fail to converge on the *citystreet*, *downtown*, and *farmland* maps. This suggests that as the visual complexity of scenes and the density of elements increase, directly applying single-stage reinforcement learning is highly challenging and unlikely to converge. These results demonstrate the effectiveness of the CBT strategy.

**Robustness under wind gusts and precipitation.** In the wind gust simulation experiments, we apply wind velocities in the range of $[2.5, 7.5]m/s$ for forward and lateral directions, and angular rate disturbances of $[0.05, 0.15]rad/s$ around the yaw axis to mimic turbulence and gusts.

### E.4 Experiments in Real-world Scenarios

We selected eight video sequences each from the VOT [30], DTB70 [35], and UAVDT [20] datasets to evaluate the transferability of GC-VAT. Specifically, from the VOT benchmark, we chose the videos *car1*, *car3*, *car6*, *car8*, *carchase*, *car16*, *following*, and *car9*. From the DTB70 benchmark, we selected *Car2*, *Car4*, *Car5*, *RcCar4*, *Car8*, *RaceCar*, *RaceCar1*, and *RcCar3*. From the UAVDT benchmark, we chose *S1603*, *S0201*, *S0101*, *S0306*, *S1201*, *S0303*, *S1301*, and *S1701*. We provide qualitative visualizations for representative video sequences. Specifically, Fig. 14 shows the output actions for a video in VOT [30] named *car6*. Fig. 15 shows the output actions for a video in VOT named *car8*. Fig. 16 shows the output actions for a video in DTB70 [35] named *Car4*. Fig. 17 shows the output actions for a video in VOT named *RaceCar1*.

## F Limitation

Although we validate the effectiveness of DAT and GC-VAT using real-world images and simple real-world scenarios, deploying the algorithm in truly open environments remains highly challenging. This is primarily due to the presence of numerous similar interfering objects and the high complexity of real-world conditions, which still exhibit a significant gap compared to simulated environments. We will further enhance the algorithm's adaptability and conduct testing in real open-world environments.

