# OpenReview forum: "Open-World Drone Active Tracking with Goal-Centered Rewards"
_NeurIPS.cc/2025/Conference — NeurIPS 2025 poster_

### Official Review · Reviewer_Vwhj · 2025-06-12

**Clarity:** 2
**Significance:** 3
**Originality:** 2
**Rating:** 5
**Confidence:** 3

**Summary:**

In this paper, the authors studied the problem of Visual Active Tracking (VAT) on drones. The authors presented a new open-world drone active air-to-ground tracking benchmark named DAT that (1) includes many different large-scale cases and can add new scenes automatically, (2) is capable of simulating full dynamics with human-like target movement, realistic setting, and can deal with both aerial and ground robots. On top of the new benchmark, a VAT algorithm called GC-VAT is proposed, which utilizes goal-centered rewards and leverages a Curriculum-Based Training strategy to achieve better tracking performance. The results are validated in experiments with different scenes.

**Questions:**

See above

**Ethical Concerns:**

["NO or VERY MINOR ethics concerns only"]

**Final Justification:**

The authors' rebuttal has resolved my concerns.

**Limitations:**

No negative societal impact need to be discussed

**Paper Formatting Concerns:**

No formatting concerns

**Quality:**

3

**Strengths And Weaknesses:**

Strength:
1. This paper is well-motivated, clearly written and organized.
2. The workload of this paper is significant. The proposed benchmark DAT is realistic and generalizable, which can contribute to the field. The proposed GC reward offered significant improvement compared with SOTA in the experiments on DAT.

Weakness:
1. The performance of GC-DAT with SOTA methods are evaluated in the proposed benchmark DAT, which contains many new properties that the experiment environments of the previous works did not have. These properties are useful, but the performance of the proposed method would be more reliable if the authors could conduct the experiments and compare the performance in those experiments, or remove the properties (like weather and complex dynamics, which are not modelled in previous works)
2. The font of the texts in the figures are too small to read
3. Although in line 298, the authors mentioned that testing the performance on real images is a method used by previous work. However, since the proposed benchmark is pretty realistic and models many properties in the real world, it would be especially interesting if GC-VAT could be tested in the real world on real robots. In this way, the significance of this work and the generalization capability of the benchmark can be better demonstrated.

---

> ### Author Rebuttal · Authors · 2025-07-31
>
> We greatly appreciate your recognition of the motivation, clarity, and significance of our work, as well as the acknowledgment of the improvement brought by our GC reward in the experiments on DAT.
>
> >Q1. The performance of GC-DAT with SOTA methods are evaluated in the proposed benchmark DAT, which contains many new properties that the experiment environments of the previous works did not have. These properties are useful, but the performance of the proposed method would be more reliable if the authors could conduct the experiments and compare the performance in those experiments, or remove the properties (like weather and complex dynamics, which are not modelled in previous works).
>
> **A1.** We appreciate the valid concern regarding benchmark fairness. We clarify the reliability of our experimental comparisons, demonstrating the superiority of our method, and strengthening our empirical validation as follows:
>
> * **Fairness of experimental comparisons.** The new properties modeled in the DAT benchmark, such as multiple weather conditions and complex visual occlusions (e.g., trees and tunnels), are not artificially designed to favor any specific method,  but are directly inspired by real-world challenges that UAV agents routinely encounter during visual active tracking. This characteristic makes DAT more aligned with real-world scenarios compared to existing benchmarks. Therefore, experimental comparisons on DAT offer a more reliable and realistic evaluation of model performance under real-world constraints.
> * **Superiority of the GC-VAT method stems from innovative reward design, independent of new properties.** The proposed goal-centered reward (Section 4.2) is inherently general-purpose: it only requires the target’s 2D image-plane projection and does not rely on task-specific heuristics. This enables accurate target localization under varying viewpoints, making it applicable to a wide range of VAT settings.
>
>     As theoretically analyzed in Section 5, the goal-centered reward accurately reflects the target position in the image plane under varying viewpoints, providing consistent policy guidance. In contrast, existing methods assume fixed forward views and use distance-based rewards that capture Euclidean proximity but not visual alignment. When viewpoints change, these rewards fail to indicate whether the target is in view or properly framed.
> * **Our method significantly outperforms baseline methods even when new properties are removed**. We simplify the DAT scenarios to fixed sunny weather conditions with targets moving only along straight roads, and compare the within-scene and cross-scene performance of GC-VAT against baseline methods, with all methods trained on the simplified citystreet-day map. As shown in Table A, even when properties such as weather variations and complex dynamics are removed, the GC-VAT method continues to significantly outperform baseline methods.
>
> Table A. Comparison result on simplified citystreet-day map.
> |Metric: CR|Within-Scene|Cross-Scene|
> |-|-|-|
> |AOT|15±1|15±0|15±0|
> |D-VAT|13±1|13±0|13±0|
> |**Ours**|**266±76**|**148±71**|
>
> |Metric: TSR|Within-Scene|Cross-Scene|
> |-|-|-|
> |AOT|0.08±0.01|0.08±0.00|0.08±0.00|
> |D-VAT|0.07±0.01|0.07±0.00|0.07±0.00|
> |**Ours** |**0.82±0.14**|**0.52±0.23**|
>
> >Q2. The font of the texts in the figures are too small to read.
>
> **A2.** Thank you for your suggestion. We have incorporated the modifications in the revised version of our manuscript.
>
> >Q3. Since the proposed benchmark is pretty realistic and models many properties in the real world, it would be especially interesting if GC-VAT could be tested in the real world on real robots.
>
> **A3.** We appreciate the reviewer’s valuable concern regarding the real-world validation of our method. We are pleased to inform that **we have successfully conducted real-world drone deployment**. Specifically:
>
> * **Real-World Drone Deployment**. We deploy the GC-VAT policy on a DJI Mini 3 Pro drone to actively track a moving vehicle. We use DJI’s Mobile SDK to stream live video to a computer for policy inference and to send control actions back to the drone for real-time execution, forming a closed-loop system.
> * **Zero-Shot Sim-to-Real Deployment and Results.** GC-VAT demonstrates strong robustness without any fine-tuning, despite real-world challenges such as airflow, sensor noise, and communication latency. As shown in Table B, across **four test trajectories**, it achieves an **average zero-shot Tracking Success Rate of 88.4%** and an **average Correct Action Rate of 81.3%**, demonstrating its strong effectiveness in real-world conditions.
>
> We acknowledge that visualizations of Sim2Real deployment would improve clarity. However, due to NeurIPS rebuttal policy, images and videos are not permitted at this stage. We will include  frames and clips in the revised manuscript.
>
> Table B. Results of real-world drone deployment. AOT [34] and D-VAT [17] fail to learn effective tracking policies even in simulation, and therefore cannot be successfully deployed on real drones.
> |Metric|Tracking Success Rate|Correct Action Rate|
> |-|-|-|
> |AOT|-|-|
> |D-VAT|-|-|
> |**Ours**|**88.4%±8.0%**|**81.3%±1.4%**|
>
> ****
> We sincerely hope our clarifications above have addressed your concerns. We would be grateful if you could kindly reconsider the evaluation of our paper.

---

### Official Review · Reviewer_92vZ · 2025-06-25

**Clarity:** 3
**Significance:** 4
**Originality:** 4
**Rating:** 5
**Confidence:** 4

**Summary:**

This paper focuses on the problem of active visual tracking of drones in an open world environment, and proposes the first relevant benchmark dataset DAT and a reinforcement learning-based method GC-VAT. DAT contains 24 urban scenes, with human-like target behavior and realistic dynamic simulation, and a digital twin tool that can generate scenes infinitely. GC-VAT designs a target center reward function to accurately feedback the target position, and also introduces a curriculum training strategy to improve performance in complex environments. Experiments show that it performs significantly better than existing methods in simulator and real image tests.

**Questions:**

1. Although DAT provides human-like behavior goals, in actual applications, the target behavior may be more complex and unpredictable. How does the GC-VAT method handle these situations?
2. In an open world environment, drones may encounter dynamic obstacles. How does the GC-VAT method handle these dynamic obstacles?
3. How to solve the problem that the curriculum training strategy may be unstable during the transition from a simple environment to a complex environment?
4. How does the reward function handle when the target is surrounded by multiple similar objects?
5. The parameter selection of domain randomization and the division of environmental complexity in the curriculum training strategy are relatively complex, and there is room for further optimization and improvement

**Ethical Concerns:**

["NO or VERY MINOR ethics concerns only"]

**Paper Formatting Concerns:**

The paper is formatted correctly

**Quality:**

4

**Strengths And Weaknesses:**

Advantages:
1. The DAT benchmark dataset proposed in the paper contains a rich variety of scenes and target types, and its effectiveness and practicality are verified through simulator and real-world image experiments. At the same time, the GC-VAT method performs well in the experiment, and has significant improvements in key indicators such as cumulative rewards compared to existing methods, proving the effectiveness of the method.
2. Starting from the actual needs of the UAV visual active tracking problem, the article systematically analyzes the shortcomings of existing methods, and then proposes the DAT benchmark dataset and GC-VAT method, and elaborates on the design ideas, implementation details and theoretical basis of the two, so that readers can clearly understand the motivation, methods and contributions of the research.
3. DAT is the first open-world UAV active tracking benchmark dataset. Its rich scenes, high-fidelity dynamics simulation and unlimited scene generation capabilities provide a powerful tool for research in the field of UAV tracking, which will help promote further development and innovation in this field.
4. The target-centered reward function and curriculum training strategy in GC-VAT are innovative. The target center reward function can accurately feedback the position of the target in the image and guide the drone to keep the target centered, solving the problem of inaccurate feedback in different perspectives of existing methods; the course training strategy effectively improves the performance of the model in complex environments.

Disadvantages:
1. Although experimental verification has been conducted on simulators and real-world images, the paper has not been fully tested on real drones. Therefore, it is impossible to fully evaluate the performance and robustness of the GC-VAT method in real environments, especially when facing complex factors such as airflow interference and sensor noise that real drones may encounter.
2. Although DAT contains a variety of urban-scale scenes, the environment in the real world is complex and changeable, and there are some special scenes or extreme cases that are not covered, resulting in insufficient generalization ability of GC-VAT in these unseen scenes, affecting the breadth of its practical application.
3. In the implementation process of GC-VAT, such as the parameter selection of domain randomization and the division of environmental complexity in the curriculum training strategy, there is room for further optimization and improvement to further improve the performance and efficiency of the method.
4. Although the target center reward function can effectively guide the drone to keep the target centered in most cases, in some complex scenarios, such as when the target is surrounded by multiple similar objects, the reward function may not be able to accurately distinguish the target object, resulting in tracking failure.
5. The curriculum training strategy may experience training instability during the transition from simple to complex environments.
6. Although DAT provides human-like behavior targets, in actual applications, target behaviors may be more complex and unpredictable. The GC-VAT method requires further training and adjustment to maintain good tracking performance when facing these unseen target behaviors.
7. In an open world environment, drones may encounter dynamic obstacles. However, the paper does not discuss in detail how the GC-VAT method handles these dynamic obstacles.
8. As task requirements change and new scenarios emerge, the GC-VAT method may need to be expanded or modified. However, the paper does not discuss in detail the scalability and modular design of the model, which may require the entire model to be redesigned and trained when facing new tracking tasks or environments, increasing the difficulty and cost of application.

---

> ### Author Rebuttal · Authors · 2025-07-31
>
> >Q1. The paper has not been fully tested on real drones.
>
> **A1.** We sincerely thank the reviewer for the insightful comment. We have **successfully conducted real drone deployment**.
> * **Real-World Drone Deployment**. We deploy the GC-VAT policy on a DJI Mini 3 Pro drone to actively track a moving vehicle. We use DJI’s Mobile SDK to stream live video to a computer for policy inference and to send control actions back to the drone for real-time execution, forming a closed-loop system.
> * **Zero-Shot Sim-to-Real Deployment and Results.** GC-VAT demonstrates strong robustness without any fine-tuning, despite real-world challenges such as airflow, sensor noise, and communication latency. As shown in Table A, across **four test trajectories**, it achieves an **average zero-shot Tracking Success Rate of 88.4%** and an **average Correct Action Rate of 81.3%**, demonstrating its strong effectiveness in real-world conditions.
>
> We acknowledge that visualizations of Sim2Real deployment would improve clarity. However, due to NeurIPS rebuttal policy, images and videos are not permitted at this stage. We will include  frames and clips in the revised manuscript.
>
> Table A. Results of real-world drone deployment. AOT [34] and D-VAT [17] fail to learn effective policies even in simulation, and therefore cannot be successfully deployed on real drones.
> |Metric|Tracking Success Rate|Correct Action Rate|
> |-|-|-|
> |AOT|-|-|
> |D-VAT|-|-|
> |**Ours**|**88.4%±8.0%**|**81.3%±1.4%**|
>
> >Q2. GC-VAT lacks sufficient generalization in unseen scenes due to uncovered special and extreme cases.
>
> **A2.** We choose dense fog as a representative extreme case. As shown in Table B, GC-VAT maintains strong performance under such conditions without retraining.
> * **Example of Extreme Cases**. According to Chinese national standards (GB/T 27964-2011) [r1], visibility below 200 meters is classified as dense fog, posing significant risks to UAV operations. We use 200m visibility as a representative extreme case.
>
> * **Additional Experiment and Results**. We train GC-VAT on daytime data and test it under 200m visibility. From Table B, GC-VAT maintains stable performance, with only a marginal drop in Tracking Success Rate (0.94→0.90), indicating reasonable generalization to extreme cases.
>
> Table B. Results on citystreet-foggy map under extreme cases.
>
> |Visibility(m)|CR|TSR|
> |-|-|-|
> |200|298±102|0.90±0.15|
> |**1000(Original)**|**316±84**|**0.94±0.14**|
>
> [r1] GB/T 27964-2011: Forecasting Grades of Fog, 2011.
>
> >Q3. The parameter selection in domain randomization and curriculum training can be further optimized.
>
> **A3.** We provide additional ablation studies to further justify our parameter selection and curriculum strategy.
>
> * **Domain randomization parameters are empirically optimal**. For domain randomization, we test height ranges: [13,22] (ours), [17,22], and [21,22]. As shown in Table C, our range [13,22] achieves the highest performance, validating our parameter selection.
> * **Curriculum design has room for improvement, but our strategy offers an efficient balance.** We compare our curriculum strategy with a three-stage variant: (1) fixed-color targets; (2) varying color targets; (3) full complexity. From Table D, it achieves slightly higher Cumulative Reward (CR) than our method in within-scene and cross-scene settings, confirming potential for improvement. Nevertheless, our approach attains strong performance with simpler design.
> * **Superior performance is clear**. Despite potential for further tuning, our method already significantly outperforms existing SOTA approaches, demonstrating the effectiveness of our current design.
>
> Table C. Ablation studies on height domain randomization.
> |Height Random/Metric: CR|Within-Scene|Cross-Scene|Cross-Domain|
> |-|-|-|-|
> |[17,22]|221±95|112±87|187±111|
> |[21,22]|174±60|122±75|147±66|
> |**[13,22] (Ours)**|**279±110**|**144±111**|**287±105**|
>
> |Height Random/Metric:TSR|Within-Scene|Cross-Scene|Cross-Domain|
> |-|-|-|-|
> |[17,22]|0.68±0.20|0.48±0.27|0.65±0.24|
> |[21,22]|0.50±0.36|0.46±0.22|0.49±0.17|
> |**[13,22] (Ours)**|**0.80±0.30**|**0.52±0.29**|**0.82±0.23**|
>
> Table D. Comparisons between the two-stage curriculum learning (Ours) and a three-stage variant
> |Metric: CR|Within-Scene|Cross-Scene|Cross-Domain|
> |-|-|-|-|
> |3-stage CL|**329±75**|**153±120**|239±112|
> |Ours|279±110|144±111|**258±110**|
>
> |Metric:TSR|Within-Scene|Cross-Scene|Cross-Domain|
> |-|-|-|-|
> |3-stage CL|**0.87±0.24**|0.51±0.32|0.72±0.29|
> |Ours|0.80±0.30|**0.52±0.29**|**0.82±0.23**|
>
> >Q4. When the target is surrounded by similar objects, the reward function may fail to distinguish it.
>
> **A4.** We respond to this concern from two perspectives: (1) the target-specific design of the reward function and (2) the empirical robustness of GC-VAT under visually similar distractors.
>
> * **Target-awareness in the reward function.** Our reward function uses the target’s ground-truth position, which is widely accepted in VAT [17,34]. Therefore, even with visually similar distractors, the reward signal reliably corresponds to the designated target.
> * **Empirical robustness under visually similar distractors.** To further evaluate GC-VAT’s robustness to distractors, we introduce a similar-looking vehicle near the target. From Table E, our model maintains high tracking performance, indicating effective discrimination of the true target despite confusing objects.
>
> Table E. Results of similar-looking vehicles on citystreet-day map.
> |Distractor Number|CR|TSR|
> |-|-|-|
> |1|263±133|0.76±0.24|
> |**0 (Original)**|**279±110**|**0.80±0.30**|
>
> >Q5. The curriculum training strategy may experience training instability.
>
> **A5.** We would like to emphasize that our Curriculum-Based Training (CBT) strategy is designed to prevent instability, and both its mechanism and empirical results confirm stable transitions without performance drops or oscillations.
>
> * **Our CBT design ensures stable transitions.** As detailed in Section 4.3 and Algorithm 1, the transitions follow a convergence criterion rather than fixed steps: training progresses only when the reward stabilizes above a threshold η. This ensures that the agent learns a robust policy before exposure to complexity (e.g., occlusions), preventing training instability.
>
> * **Training is smooth and monotonic.** Figure 5 in the paper shows the reward curves rise steadily across all scenes, with no drop or oscillation at the curriculum transition point. This demonstrates a stable and seamless curriculum transition.
>
> >Q6. The GC-VAT method requires further training when facing these unseen target behaviors.
>
> **A6.** We would like to clarify that GC-VAT, trained on the diverse behaviors in DAT, inherently generalizes to unseen target behaviors without requiring further training.
>
> * **Robustness under Rare or Extreme Behaviors.** As mentioned in Section 6.4, we perform real-image experiments on the RaceCar1 video from DTB70 dataset, featuring extreme target behaviors like drifting and serpentine motion. Table 5 in the paper shows GC-VAT achieves zero-shot generalization to unseen behaviors with a **Correct Action Rate of 0.880**.
> * **DAT Captures Common Real-World Target Behaviors.** As shown in Figure 2, DAT includes diverse target behaviors, such as lane changes. These maneuvers cover the majority of real-world driving behaviors [r2]. This diversity enables GC-VAT to learn robust policies that generalize to unseen behaviors without further training.
>
> [r2] The highd dataset: A drone dataset of naturalistic vehicle trajectories on german highways for validation of highly automated driving systems, ITSC 2018.
>
> >Q7. The paper does not discuss in detail how the GC-VAT method handles the dynamic obstacles.
>
> **A7.** GC-VAT method handles dynamic obstacles through collision-aware reset mechanism.
>
> * **Collision-Aware Reset Mechanism**. Our training framework encourages the drone to avoid both static (buildings, trees) and dynamic obstacles (drones, vehicles). If a collision occurs during training, the episode is immediately reset with a strong penalty. Therefore, the drone learns visual obstacle avoidance for higher rewards.
>
> >Q8. Insufficient discussion on model scalability and modular design limits adaptability to new tasks or environments.
>
> **A8.** We clarify that GC-VAT is scalable and generalizable, as evidenced by: (1) a general-purpose reward function, (2) a model-agnostic curriculum strategy, and (3) empirical generalization to unseen targets and environments.
> * **General-Purpose Reward Design.** The proposed goal-centered reward function (detailed in section 4.2) is applicable to any VAT setting that involves viewpoint variation and image-based tracking. It only requires the target’s image-plane projection and no task-specific heuristics, enabling accurate target localization under varying viewpoints.
>
> * **Model-Agnostic Curriculum Training.** Our curriculum learning strategy is independent of GC-VAT’s architecture. It can be easily applied to any VAT policy, as it relies only on the agent’s performance (e.g., average reward) to adaptively schedule training difficulty. This design enhances agent performance and accelerates learning in complex environments.
>
> * **Empirical Validation of Generalization.** From Table F, our model maintains strong performance when facing an unseen target (Bus). Moreover, Tables 2 and 3 in the paper show that GC-VAT achieves cross-scene and cross-domain transferability. These results show that our method can adapt to new tasks and environments.
>
> Table F. Results of unseen targets (Bus).
> |Metric: CR|Within-Scene|Cross-Scene|Cross-Domain|
> |-|-|-|-|
> |Target: Bus (unseen)|229±92|131±89|207±94|
> |**Ours**|**279±110**|**144±111**|**258±110**|
>
> |Metric: TSR|Within-Scene|Cross-Scene|Cross-Domain|
> |-|-|-|-|
> |Target: Bus (unseen)|0.79±0.25|0.50±0.33|0.79±0.27|
> |**Ours**|**0.80±0.30**|**0.52±0.29**|**0.82±0.23**|
>
> ****
> We sincerely hope our clarifications above have addressed your concerns.

---

### Official Review · Reviewer_e5dU · 2025-07-03

**Clarity:** 2
**Significance:** 2
**Originality:** 3
**Rating:** 4
**Confidence:** 4

**Summary:**

This paper presents GC-VAT, a comprehensive reinforcement learning solution with goal-centered rewards for open-world drone active tracking. The authors develop DAT, the first benchmark for drone active air-to-ground tracking, featuring 24 city-scale scenes with high-fidelity dynamics simulation and tools for generating human-like target behaviors and unlimited scenes. To enhance tracking in complex environments, they design a Goal-Centered Reward function providing precise viewpoint feedback and a Curriculum-Based Training strategy for progressive agent adaptation. Experiments on simulator and real-world images demonstrate GC-VAT achieves 400% improvement over state-of-the-art methods in cumulative reward metrics, validating the method's robustness and generalization across diverse dynamic scenarios.

**Questions:**

1. The paper only evaluates the model on the VOT and DTB70 datasets and draws conclusions about its effectiveness in real-world images based on the average correct action rate from just five videos, which presents certain limitations. To strengthen the persuasiveness of the conclusion, the paper should expand the test dataset range, report the average correct action rates for each dataset individually, and demonstrate the model's generalizability by showing consistent performance across multiple datasets.

2. The complexity of some of the scene constructions is insufficient. For example, the village map in the paper presents mountainous terrain with tunnels, but the road surface remains flat and unaffected by the terrain, making the distinction between mountainous and flat areas minimal. It is recommended to increase the scene’s complexity, particularly with regard to obstacles and terrain features.

3. The paper provides limited discussion on common external disturbances in real-world environments, such as gusts, wind turbulence, and the attenuation effects of precipitation on sensors. It is recommended to include further analysis or experimental validation on these aspects.

4. The arrow in Figure 13 should be explained in the following text to help with understanding.

5. It is recommended to add a color bar to the reward contour plot in Figure 4 to help readers better understand the value range.

**Ethical Concerns:**

["NO or VERY MINOR ethics concerns only"]

**Final Justification:**

Most of my concerns have been addressed.

**Limitations:**

yes

**Quality:**

2

**Strengths And Weaknesses:**

Strengths

1. The DAT proposed in this paper is the first open-world benchmark for drone active aerial-to-ground tracking. It includes 24 city-scale scenes, featuring high-fidelity dynamic simulations and human behavior models, providing a comprehensive and realistic testing environment for algorithm validation.
2. By designing a Goal-Centered Reward function and a Curriculum-Based Training strategy, GC-VAT significantly improves tracking performance in complex dynamic environments. Experiments show that its cumulative reward metric is approximately 400% higher than existing methods.
3. The paper not only proposes a new reward design approach but also validates its effectiveness through theoretical analysis and experimental verification. Particularly, its ability to provide precise feedback from different perspectives enhances the credibility of the method.

Weaknesses

1. The paper tests the model only on the VOT and DTB70 datasets, and its effectiveness in real-world images is evaluated based on the average correct action rate from only a small number of videos, which raises concerns about the generalizability of the conclusions.
2. The experimental evaluation lacks consideration of real-world environmental challenges, including both external disturbances (e.g., wind gusts, turbulence, and precipitation effects on sensors) and realistic terrain complexity (e.g., overly simplified village map with flat roads), potentially compromising the method's practical applicability.

---

> ### Author Rebuttal · Authors · 2025-07-31
>
> We sincerely appreciate your recognition of our work in developing the first open-world benchmark for drone active aerial-to-ground tracking, as well as the acknowledgment of our innovative reward design and training strategy. Your feedback on the effectiveness and credibility of our approach is truly valuable to us.
>
> >Q1. The paper should expand the test dataset range, report the average correct action rates for each dataset individually.
>
> **A1.** We thank the reviewer for the valuable feedback. We have extended our evaluation as suggested on real-world images and successfully **deployed the GC-VAT policy on a physical DJI Mini 3 Pro drone** to validate its consistent tracking capability in real-world scenarios. This real-world deployment demonstrates the practical effectiveness and robustness of our method beyond simulation.
>
> * **Expanded Real-World Image Evaluations**. We have expanded the number of test videos on real-world image datasets. Specifically, we now evaluate zero-shot transfer performance on 8 videos from VOT [26], 8 from DTB70 [30], and 8 from UAVDT [r1], expanding the original evaluation scale. The results in Table A-C show consistently strong performance across all datasets:
>     * **VOT**: Average Correct Action Rate (CAR) = **0.795**
>     * **DTB70**: Average CAR = **0.833**
>     * **UAVDT**: Average CAR = **0.802**
>
>     This consistent performance across diverse datasets with varying environmental conditions, camera motions, and target dynamics confirms the strong generalization ability of GC-VAT.
>
> * **Real-World Drone Deployment**. Furthermore, as a critical step beyond image-based evaluation, we deploy the GC-VAT policy on a DJI Mini 3 Pro drone for real-world vehicle tracking. GC-VAT demonstrates strong robustness without any fine-tuning, despite real-world challenges such as airflow, sensor noise, and communication latency. As shown in Table D, across **four test trajectories**, GC-VAT achieves an **average zero-shot Tracking Success Rate of 88.4%** and an **average Correct Action Rate of 81.3%**. This successful zero-shot Sim-to-Real transfer validates the realism of the DAT benchmark and the practical applicability of our approach.
>
>     We acknowledge that visualizations of Sim2Real deployment would improve clarity. However, due to NeurIPS rebuttal policy, images and videos are not permitted at this stage. We will include  frames and clips in the revised manuscript.
>
>
> Table A. Results (metric is Correct Action Rate) of 8 videos in VOT benchmark.
> |Video|car1|car3|car6|car8|carchase|car16|following|car9|
> |-|-|-|-|-|-|-|-|-|
> |Random|0.418|0.434|0.418|0.430|0.429|0.421|0.314|0.439|
> |**Ours**|**0.696**|**0.845**|**0.754**|**0.833**|**0.870**|**0.834**|**0.773**|**0.756**|
>
> Table B. Results (metric is Correct Action Rate) of 8 videos in DTB70 benchmark.
> |Video|Car2|Car4|Car5|RcCar4|Car8|RaceCar|RaceCar1|RcCar3|
> |---|---|---|---|---|---|---|---|---|
> |Random|0.419|0.421|0.429|0.436|0.411|0.462|0.430|0.400|
> |**Ours**|**0.757**|**0.894**|**0.893**|**0.876**|**0.803**|**0.713**|**0.880**|**0.851**|
>
> Table C. Results (metric is Correct Action Rate) of 8 videos in UAVDT [r1] benchmark.
> |Video|S1603|S0201|S0101|S0306|S1201|S0303|S1301|S1701|
> |---|---|---|---|---|---|---|---|---|
> |Random|0.435|0.438|0.422|0.437|0.445|0.385|0.397|0.407|
> |**Ours**|**0.896**|**0.806**|**0.865**|**0.773**|**0.867**|**0.735**|**0.760**|**0.713**|
>
> Table D. Results of real-world drone deployment. AOT [34] and D-VAT [17] fail to learn effective policies even in simulation, and therefore cannot be successfully deployed on real drones.
> |Metric|Tracking Success Rate|Correct Action Rate|
> |-|-|-|
> |AOT|-|-|
> |D-VAT|-|-|
> |**Ours**|**88.4%±8.0%**|**81.3%±1.4%**|
>
> [r1] The unmanned aerial vehicle benchmark: Object detection and tracking, ECCV 2018.
>
> >Q2. The village map in the paper presents mountainous terrain with tunnels, but the road surface remains flat and unaffected by the terrain, making the distinction between mountainous and flat areas minimal. It is recommended to increase the scene’s complexity, particularly with regard to obstacles and terrain features.
>
> **A2.** To directly address the reviewer’s suggestion, we leverage DAT’s digital twin tool to upgrade terrain realism, and validate GC-VAT’s zero-shot generalization capability on unseen terrains.
>
> * **Enhancing Terrain Realism with the Digital Twin Tool.** DAT benchmark includes a **digital twin tool** (detailed in Section 3.1 and Appendix B) that supports the integration of real-world elevation data, enabling the generation of scenes with realistic, elevation-aware terrain and road surfaces. In response to the reviewer’s suggestion, we will enhance the benchmark in the revised manuscript by explicitly incorporating terrain complexity metrics and providing updated scenes with sloped roads and elevation variations. We believe this will further strengthen the realism and utility of DAT for future research.
>
> * **Generalization to Real-World Terrain Variations**. Notably, GC-VAT, trained on flat roads in simulation, achieves zero-shot generalization to real-world scenarios with significant elevation changes and sloped terrains. This is enabled by the goal-centered reward (in Section 4.2), which accurately reflects the target’s position in the image and decouples policy learning from specific terrain geometry. As shown in Table 5 in the paper, GC-VAT achieves a **Correct Action Rate of 0.880** on the challenging RaceCar1 sequence in the DTB70 dataset, where the target moves on a track with pronounced slopes and elevation variations.
>
>
> >Q3. The paper provides limited discussion on common external disturbances in real-world environments, such as gusts, wind turbulence, and the attenuation effects of precipitation on sensors. It is recommended to include further analysis or experimental validation on these aspects.
>
> **A3.** We conduct rigorous tests under wind gusts and sensor degradation caused by precipitation. Results in Table E and Table F demonstrate that GC-VAT maintains stable performance under the interference.
>
> * **Resilience to Wind Perturbations.** To simulate realistic wind effects, we introduce randomized perturbations along the forward, lateral, and yaw directions during testing. Specifically, we apply wind velocities in the range of [2.5, 7.5] m/s for forward and lateral directions, and angular rate disturbances of [0.05, 0.15] rad/s around the yaw axis to mimic turbulence and gusts. The results under such wind disturbances are summarized in Table E, where the model is trained on citystreet-day and evaluated on citystreet-foggy with added wind perturbations. The Tracking Success Rate (TSR) drops by less than 0.06, demonstrating that GC-VAT maintains strong robustness under significant wind disturbances.
> * **Visual Robustness under Precipitation.** Since our method relies solely on monocular RGB images for decision making, precipitation primarily affects tracking through two mechanisms: visibility attenuation and blurring caused by raindrops.
>     * **Visibility Attenuation:** The robustness of our method under fog-induced visibility degradation is already demonstrated in existing experiments, such as the cross-domain evaluation on citystreet-foggy. As shown in Table 2 and Table 3 of the paper, GC-VAT achieves a tracking success rate (TSR) with only a 0.04 drop (0.80→0.76) when transferring from clear to foggy conditions. This small performance gap indicates that **GC-VAT effectively adapts to visibility attenuation caused by precipitation-related fog**.
>     * **Blurring Caused by Raindrops:** We follow established practices in test-time adaptation literature [r2] to simulate realistic rainfall effects. Specifically, we train the policy on citystreet-day map and evaluate under synthetically generated rain in within-scene, cross-scene, and cross-domain settings. To ensure realism, we exclude snowy conditions from the cross-domain evaluation, as snow and rain rarely co-occur in real-world environments. The results in Table F show only marginal performance degradation (less than 0.07 in Tracking Success Rate) under rain simulation, confirming that **GC-VAT is robust to blurring caused by raindrops**.
>
> Table E. Performance under wind disturbances and clean conditions.
> |Exp_Name|CR|TSR|
> |-|-|-|
> |w/ Forward|302±94|0.91±0.18|
> |w/ Lateral|304±82|0.91±0.19|
> |w/ Yaw|301±120|0.88±0.23|
> |**w/o Wind**|**316±84**|**0.94±0.14**|
>
> Table F. Performance under rainy and clean conditions.
> |Metric: CR|Within-Scene|Cross-Scene|Cross-Domain|
> |-|-|-|-|
> |w/ rain|266±110|139±109|274±103|
> |**w/o rain**|**279±110**|**144±111**|**287±105**|
>
> |Metric: TSR|Within-Scene|Cross-Scene|Cross-Domain|
> |-|-|-|-|
> |w/ rain|0.74±0.29|0.45±0.30|0.77±0.29|
> |**w/o rain**|**0.80±0.30**|**0.52±0.29**|**0.83±0.22**|
>
> [r2] Heavy rain image restoration: Integrating physics model and conditional adversarial learning, CVPR 2019.
>
> >Q4. The arrow in Figure 13 should be explained in the following text to help with understanding.
>
> **A4.** In Figure 13, the arrow indicates the visualization of data points. For instance, the yellow point in the upper right corner represents a target point located to the right of the image center, while the arrow points to the corresponding actual image.
>
> >Q5. It is recommended to add a color bar to the reward contour plot in Figure 4 to help readers better understand the value range.
>
> **A5.** Thank you for your suggestion. We have incorporated the modifications in the revised version of our manuscript.
>
> ****
> We sincerely hope our clarifications above have addressed your concerns. We would be grateful if you could kindly reconsider the evaluation of our paper.

---

> > ### Comment · Reviewer_e5dU · 2025-08-04
> >
> > Thank you for the authors' response. Most of my concerns have been addressed. I will raise my score.

---

> > > ### Author Response · Authors · 2025-08-04
> > >
> > > Thank you for taking the time to revisit our responses and for considering raising the score. We're glad the additional experiments and clarifications addressed your concerns.

---

> ### Author Response · Authors · 2025-08-04
>
> Dear Reviewer,
>
> Thank you for your valuable feedback and thoughtful comments on our manuscript. We have submitted a point-by-point rebuttal, providing detailed responses to each of your concerns.
>
> We would like to reiterate the core innovations of our work, which we believe make significant contributions to the field:
>
> 1. **DAT: Pioneering Open-World Drone Tracking Benchmark**
>
>     * We establish the first open-world benchmark for drone active tracking featuring 24 city-scale scenes with realistic dynamics and human-like target behaviors.
>     * Our digital twin tools enable automatic scene generation from real-world geographic data, solving scalability limitations of manual environment design.
>
> 2. **GC-VAT: Theory-Driven Tracking Framework**
>     * We develop a novel reward mechanism using goal-centered image projection, formally proven to maintain effectiveness during viewpoint changes.
>     * We implement curriculum-based training that progressively transitions agents from simple to complex dynamic environments, significantly enhancing robustness.
>     * GC-VAT demonstrates approximately **400%** Cumulative Reward improvement over SOTA while maintaining consistent performance under dense fog (200m visibility), strong wind gusts, and heavy rain conditions.
>
> 3. **Validated Real-World Performance**
>     * GC-VAT achieves **88.4%** real-world Tracking Success Rate through zero-shot sim-to-real transfer on commercial drone platforms (DJI Mini 3 Pro).
>     * GC-VAT maintains **83.3%** Correct Action Rate on real-world image dataset DTB70 against challenging distractors and terrain variations.
>
> We are confident these contributions will advance the field and welcome further clarification to support your final decision.
>
> Sincerely,
> The Authors

---

### Official Review · Reviewer_CwYq · 2025-07-12

**Clarity:** 3
**Significance:** 3
**Originality:** 3
**Rating:** 4
**Confidence:** 3

**Summary:**

This paper presents DAT, a novel benchmark designed for open-world drone active air-to-ground tracking. The benchmark features high-fidelity simulation environments and human-like target behaviors, addressing the current lack of unified benchmarks and diverse environmental conditions in the drone tracking domain. In addition, the authors propose a goal-centered reward formulation that provides viewpoint-sensitive feedback to the agent.

**Questions:**

See the weakness. I think this paper to be a valuable contribution to the field of embodied AI and active perception. The new benchmark is timely and will likely benefit future research. The goal-centered reward is both novel and effective. Despite some limitations in baseline diversity and architecture choices, the paper demonstrates clear innovation, solid implementation, and meaningful insights. I recommend acceptance.

**Ethical Concerns:**

["NO or VERY MINOR ethics concerns only"]

**Limitations:**

Yes

**Quality:**

3

**Strengths And Weaknesses:**

Strengths

1.This paper is well-written and easy to follow. The motivation is clear. in current drone tracking setups, it lacks of environmental diversity and insufficient dynamic complexity. To overcome these limitation, this paper propose a new benchmark and a new method.

2.The task of drone active tracking is well-define. Figure 2 effectively illustrates the attributes and diversity of the proposed benchmark.

3.The proposed goal-centered reward is interesting. It enables the agent to reason about its viewpoints actively, which is essential in open-world scenarios. Extensive experiments valid the effectiveness of proposed components.

Weaknesses

1.The experiments include only two baseline methods. For a newly proposed benchmark to be impactful, a broader set of comparative evaluations, especially including more recent or state-of-the-art methods from the past 1-2 years would strengthen the empirical validation.

2.The current implementation uses a CNN-based backbone. To further strengthen the empirical evaluation, I suggest to include comparisons with recent transformer-based architectures.

---

> ### Author Rebuttal · Authors · 2025-07-31
>
> We sincerely thank the reviewer for the thoughtful and constructive feedback, as well as for recognizing the contributions of our work. We greatly appreciate the reviewer’s acknowledgment of the novelty and impact of the DAT benchmark, the effectiveness of the GC-VAT method, and the rigor of our theoretical and experimental analysis. This encouragement motivates us to further advance research in open-world drone active tracking.
>
> >Q1. Include more recent or state-of-the-art methods from the past 1-2 years.
>
> **A1.** We sincerely thank the reviewer for the constructive suggestion. To validate our method against the latest advancements as suggested, we include two recent VAT methods published in ECCV 2024 [r1] and RAL 2024 [r2] under the same protocol. As shown in Table A, **GC-VAT consistently outperforms the baselines** across Tracking Success Rate (TSR) and Cumulative Reward (CR) metrics. The details are as follows:
>
> * **Additional Recent VAT Methods.** To strengthen the empirical evaluation as suggested, we further include two recent methods:
>     * **Offline RL (ECCV 2024)** [r1] integrates visual foundation models with offline reinforcement learning to enable data-efficient visual active tracking in embodied agents.
>     * **FAn (IEEE RAL 2024)** [r2], a pipeline-based approach that integrates visual tracking with PID control for improved stability in drone active tracking.
> * **Updated Empirical Evaluations.** These two methods reflect the most recent developments in VAT field. The extended comparisons in Table A show that GC-VAT significantly outperforms all the baselines across within-scene, cross-scene, and cross-domain settings.
> * **Original Selection of SOTA Methods.** In the original submission, we selected two widely recognized state-of-the-art methods in VAT: AOT [34] and D-VAT [17], with D-VAT representing the most recent and highest-performing approach prior to our work.
>
>
> Table A. Comparison between Offline RL and our method on DAT.
>
> | Metric: CR|Within-Scene↑|Cross-Scene↑|Cross-Domain↑|
> |-|-|-|-|
> |Offline RL [r1]|35±5|37±8|35±4|
> |FAn [r2]|28±3|31±5|33±8|
> |**Ours**|**279±110**|**144±111**|**258±110**|
>
> | Metric: TSR|Within-Scene↑|Cross-Scene↑|Cross-Domain↑|
> |-|-|-|-|
> |Offline RL [r1]|0.09±0.01|0.10±0.02|0.09±0.01|
> |FAn [r2]|0.16±0.02|0.16±0.01|0.18±0.05|
> |**Ours**|**0.80±0.30**|**0.52±0.29**|**0.82±0.23**|
>
>
>
> [r1] Empowering Embodied Visual Tracking with Visual Foundation Models and Offline RL, ECCV 2024.
>
> [r2] Follow anything: Open-set detection, tracking, and following in real-time, RAL 2024.
>
> >Q2. Include comparisons with recent transformer-based architectures.
>
> **A2.** We have incorporated experiments comparing our CNN-based method with MobileViT [r3]. Our results indicate that the MobileViT-based method underperforms the adopted CNN-based approach in tracking performance and inference efficiency during reinforcement learning training.
>
> * **Comparisons with Lightweight Transformer.** To directly address the reviewer’s suggestion, we conduct additional experiments incorporating a MobileViT [r3] backbone into our evaluation. We choose MobileViT specifically for its focus on efficiency and edge-device applicability. As detailed in Table B and Table C, the adopted CNN-based model achieves a **65% higher average Complete Rate (CR)** within-scene compared to MobileViT. Crucially, it also delivers **6.6x faster inference speed**. In our experiments on DAT, we observe that MobileViT exhibits slower convergence and higher variance in the reward curve. This indicates challenges in learning stable and optimal policies, aligning with observations in [r4] regarding training instability in similar RL settings.
> * **The Necessity of CNNs for Edge Deployment in Visual Active Tracking.** Drone-based visual active tracking should operate on resource-constrained edge devices (e.g., onboard computers or embedded systems). This environment demands extreme efficiency: minimal computational cost, low memory footprint, and real-time inference latency are non-negotiable for reliable and safe operation. CNN architectures are the established default backbone precisely because they excel in meeting these critical constraints: they are computationally efficient, inherently lightweight, and deliver the fast inference essential for real-time tracking. This fundamental alignment with deployment realities is why CNNs dominate VAT systems [r5, r6].
>
> Table B. Comparisons between transformer-based and CNN-based model on Citystreet-day map of DAT.
> | Metric: CR|Within-Scene↑|Cross-Scene↑|Cross-Domain↑|
> |-|-|-|-|
> |MobileVit[r3]|169±73|47±21|45±25|
> |**CNN**|**279±110**|**144±111**|**258±110**|
>
> | Metric: TSR|Within-Scene↑|Cross-Scene↑|Cross-Domain↑|
> |-|-|-|-|
> |MobileVit[r3]|0.35±0.21|0.11±0.04|0.15±0.12|
> |**CNN**|**0.80±0.30**|**0.52±0.29**|**0.82±0.23**|
>
> Table C. Results of inference time on an RTX 3090 GPU
>
> |Architecture|Inference Time (s)↓|
> |-|-|
> |MobileVit[r3]|0.0106|
> |**CNN**|**0.0016**|
>
> [r3] Mobilevit: light-weight, general-purpose, and mobile-friendly vision transformer, ICLR 2022.
>
> [r4] On transforming reinforcement learning with transformers: The development trajectory, TPAMI 2024.
>
> [r5] D-vat: End-to-end visual active tracking for micro aerial vehicles, RAL 2024.
>
> [r6] End-to-end active object tracking and its real-world deployment via reinforcement learning, TPAMI 2019.
>
>
> ****
> We sincerely hope our clarifications above have addressed your concerns. We would be grateful if you could kindly reconsider the evaluation of our paper.

---

### Decision · Program_Chairs · 2025-09-17

**Decision:**

Accept (poster)

**Comment:**

In the paper the authors present a novel benchmark designed for open-world drone active air-to-ground tracking. The benchmark features high-fidelity simulation environments and human-like target behaviors. This adress the current lack of unified benchmarks and diverse environmental conditions in the drone tracking domain. The authors also propose a goal-centered reward formulation that provides viewpoint-sensitive feedback to the agent.
Strenghts of the paper include (i) first open-world benchmark for drone active aerial-to-ground tracking, (ii) well written paper, (iii) interesting goal-centered reward formulation, (iv) Good improvement to cumulative reward metric, (v) rich variety of scenes and target types, (vi) solid analysis on the shortcomings of existing methods.
Weaknesses include (i) limited experimental validation using 2 previous methods, (ii)
Reviewers also suggested replacing the CNN backbone to a transformer based one and also provided other ideas of improving the paper.
The authors provided a good rebuttal. Both reviewers and authors engaged in the discussion, after which the reviewers all suggest accepting the paper with grades 4,4,5,5. I recommend accepting the paper.